# Generating biomembrane-like local curvature in polymersomes via dynamic polymer insertion

Jiawei Sun[1], Sjoerd J. Rijpkema[1], Jiabin Luan[1], Shaohua Zhang[1] & Daniela A. Wilson [1✉]

Biomembrane curvature formation has long been observed to be essential in the change of membrane morphology and intracellular processes. The significant importance of curvature formation has attracted scientists from different backgrounds to study it. Although magnificent progress has been achieved using liposome models, the instability of these models restrict further exploration. Here, we report a new approach to mimic biomembrane curvature formation using polymersomes as a model, and poly(N-isopropylacrylamide) to induce the local curvature based on its co-nonsolvency phenomenon. Curvatures form when poly(N-isopropylacrylamide) becomes hydrophobic and inserts into the membrane through solvent addition. The insertion area can be fine-tuned by adjusting the poly(N-isopropylacrylamide) concentration, accompanied by the formation of new polymersome-based non-axisymmetric shapes. Moreover, a systematic view of curvature formation is provided through investigation of the segregation, local distribution and dissociation of inserted poly(N-isopropylacrylamide). This strategy successfully mimicks biomembrane curvature formation in polymersomes and a detailed observation of the insertion can be beneficial for a further understanding of the curvature formation process. Furthermore, polymer insertion induced shape changing could open up new routes for the design of non-axisymmetric nanocarriers and nanomachines to enrich the boundless possibilities of nanotechnology.

[1] Institute for Molecules and Materials, Radboud University, Nijmegen, the Netherlands. ✉email: d.wilson@science.ru.nl

In nature, biomembrane curvature formation is observed in fundamental cellular processes including endo- and exo-cytosis, immune response and cell motion[1]. Specific lipids and proteins[2] such as Bin/Amphiphysin/Rvs-domain proteins[3] are recruited during the processes. Notably, amphipathic helices are included in these proteins, and insertion of their hydrophobic regions into the membrane matrix to change the membrane composition is suggested to be a potent method for generating local curvature[4]. To study the curvature formation, several models have been developed, such as bilayer structures based on lipids known as liposomes[5] and synthetic molecules known as polymersomes[6–8]. In liposome models, the importance of the change of the membrane composition has been indicated by the membrane curvature formation caused by protein interaction[9–12] and peptide insertion[13], consistent with the natural processes mentioned earlier. Alterations in the composition of the membrane leads to a difference between the surface area of the outer and inner monolayer of a bilayer membrane. This so-called reduced monolayer area difference ($\Delta a$) (Supplementary Fig. 1) can influence the membrane morphology dramatically[14–17]. In spite of these remarkable advances, liposomes were found to be unstable, sensitive and difficult to be modified, which hinders their role in further exploration of biomembrane mimicking. In comparison, polymersomes have higher stability, structural versatility and surface modifiability, making them a popular tool for mimicry[8]. However, the rigidity of the polymersome membrane has limited the possibilities in shape transformation to mostly tubular or stomatocyte-like structures, since there is hardly any membrane composition change in the bilayer[7,18–22]. This not only restricted the potential to mimic biomembranes, but also limits the exploitation of polymersomes as nano-systems in different fields. In order to bring the research one step closer to practical applications, strategies which are capable of changing the composition of the membrane of polymersomes are critical.

In this study, we aim to mimic the biomembrane curvature formation using polymersomes as our model. Furthermore, we propose to use carboxylic acid terminated poly(*N*-iso-propylacrylamide) (PNIPAm) to interact with the membrane of polymersomes. Since polymersomes have glassy membranes, swelling of the membrane under addition of organic solvents allows them to obtain enough mobility and permeability to respond to environmental changes[20,23,24]. PNIPAm is well-known for its transition from a coiled hydrophilic structure to an entangled hydrophobic structure when heated above its lower critical solution temperature (LCST)[25]. Besides this, the LCST of PNIPAm can be shifted when water is mixed with specific solvents such as methanol and tetrahydrofuran (THF) due to the local interactions between the polymer chains and the solvents, known as the co-nonsolvency phenomenon[26–28]. Here, plasticizing solvent is added to rigid polymersomes to untie the entangled bilayer structure and increase the membrane flexibility and fluidity[23,24]. Consequently, the LCST of PNIPAm is decreased because of the co-nonsolvency phenomenon and an amphipathic helice-like structure is formed which can insert into the flexible membrane (Fig. 1a, b). This strategy allows us to successfully mimic the biomembrane curvature formation. Increasing the PNIPAm concentration thereby results in an increase of the insertion area (Fig. 1d). The interaction mechanism, PNIPAm segregation, local distribution and dissociation in the polymersome membrane have been investigated (Fig. 1b, c, e). These results demonstrate the capabilities of polymersomes as a biomembrane mimicking model with varying membrane compositions and allows for the investigation of membrane insertion mechanism and curvature formation. Meanwhile, polymer insertion induced shape changing would open up routes for the design of nonaxisymmetric nanocarriers,

nanomachines etc, to enrich the limitless possibilities of nanotechnology.

## Results

**Curvature formation of polymersomes via PNIPAm insertion.** We investigated the interaction between the polymersome membrane and PNIPAm by adding different amounts of PNIPAm varying from 5 to 200 µg to the polymersome solutions (Fig. 2a). PNIPAm, which is in the hydrophilic state, interacts only with the hydrophilic PEG layer of the polymersome membrane and we do not expect any other interactions at this point. A mixture of tetrahydrofuran (THF) and 1,4-dioxane (4:1 v/v,) which has a plasticizing effect on polystyrene (PS), was selected to increase the flexibility and fluidity of membrane. Concomitantly, as described before, a change in the LCST of PNIPAm and formation of a hydrophobic motif is expected in presence of the solvent. We followed changes in the transmittance during the increase of the volume fraction of co-nonsolvent (VFC) using UV–VIS spectroscopy (Fig. 2b) which confirmed the transition of PNIPAm. The LCST decreased from 32.9 °C to 19.9 °C during the addition of the organic solvent up to a VFC of 23.08%.Keeping the temperature constant at 21 °C, PNIPAm becomes gradually more hydrophobic during the solvent addition. Due to the high Tg of the PS domains, the structures could be trapped by quenching in water after solvent addition. This is an advantage of a polymersome-based membrane model over liposomal membrane models where the fluidity of the phospholipid bilayer membrane prevents trapping of the structure[8,29]. Samples were examined by TEM, SEM (Fig. 2d–i) and cryo-TEM (Supplementary Fig. 2), respectively. Interestingly, the addition of plasticizer can not only increase the flexibility, but also stretch the membrane and create space between the assembled polymers for the PNIPAm insertion. Curvature formation is observed after a point where PNIPAm becomes hydrophobic and the membrane opens up for PNIPAm insertion. The first change was observed when the PNIPAm concentration reached 10 µg. More tentacles (from 2 to 5) were formed with the increase of PNIPAm (from 25 µg to 200 µg) (Fig. 2c), as more PNIPAm was present near the membrane. We assume that the PNIPAm insertion might already take place under these conditions due to the hydrophobic effect. The process of squeezing out water and PNIPAms hydrophilic-hydrophobic transition might push them towards the hydrophobic PS chains, which deepens their insertion into the membrane. Moreover, the insertion only happened in the outer layer of the bilayer, since PNIPAm was only present in the outer solution but not the inner compartment of the polymersomes. This results in a change of $\Delta a$[15,28,30], leading to the formation of a local positive curvature. In fact, similar processes were observed in simulation studies of protein membranes interactions, where isolated N-terminal amphipathic helices are unfolded in solution but folded during insertion into a lipid membrane[31]. Here, the amphipathic transition of PNIPAm provided experimental evidence for this hypothesis.

Furthermore, 3D views of these structures were analyzed by cyro-SEM. The morphology of polymersomes changed from spherical shape to discocyte-like shape after addition of 23.08% organic solvent (Fig. 2d3), indicating a volume reduction of the polymersome[32] induced by the exchange of water with organic solvent between the inner and outer side of the membranes[20]. Furthermore, the subsequent curvature formation is based on the flattening and widening of the vesicular structure (Fig. 2e3-i3). Since the rim of this vesicular structure is more stretched and has the largest mean curvature compared to the rest of the membrane[33], more space is available here between the entangled polymer chains for the insertion of PNIPAm.

In natural membranes, amphipathic helices would preferably insert into lipid packing defects, which might have a similar effect to the flat disc membrane rim on membrane insertion, while the

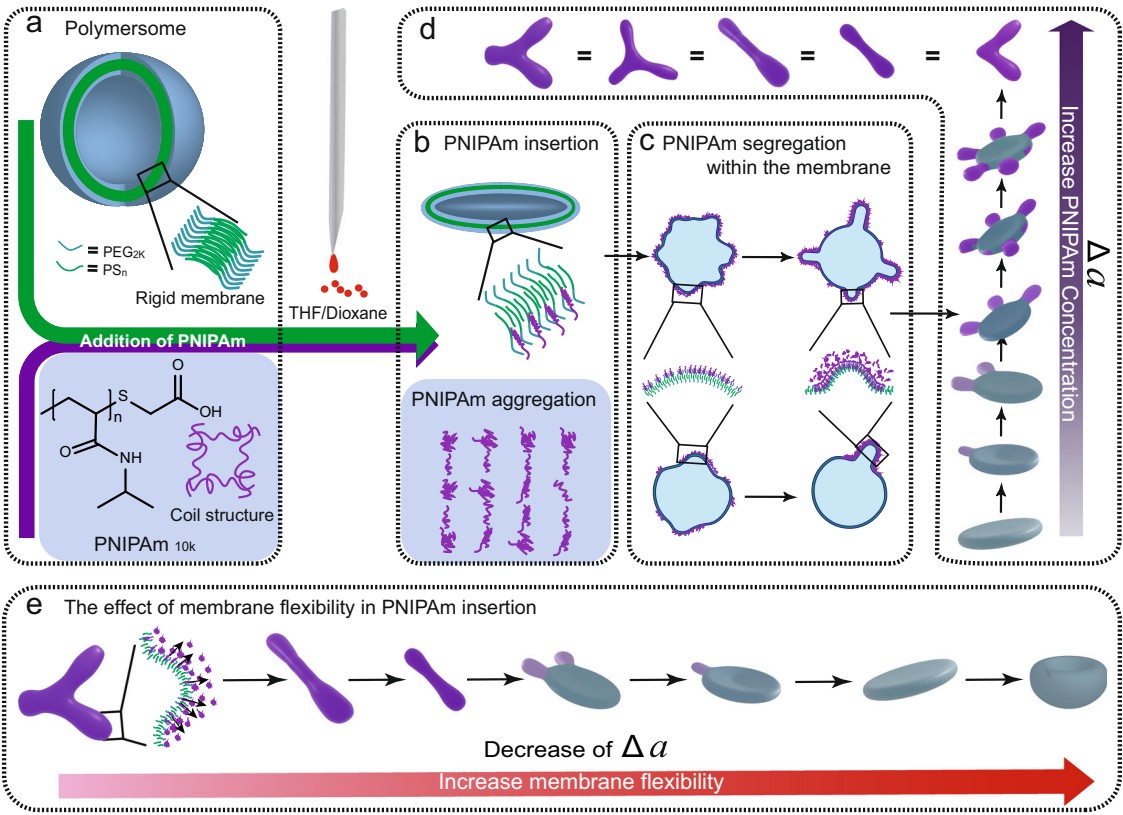

**Fig. 1 Overview of the curvature formation in polymersome model. a** Scheme of poly(ethylene glycol)-polystyrene (PEG-*b*-PS) polymersome, the membrane and PNIPAm structure. **b** Schematic representation of PNIPAm-membrane interaction after increasing the membrane flexibility through plasticizing solvent addition (23.08% v/v). The addition of solvent leads to PNIPAm insertion into polymersomes membrane due to the shift from hydrophilicity to hydrophobicity. **c** Scheme of PNIPAm rearrangement in the polymersome membrane in different PNIPAm concentrations. **d** By increasing the PNIPAm concentration 3D structures of different shapes were obtained, including polymersomes with tentacles, cigar-like shapes, rackets-like shapes, boomerang-like shapes and 3-armed starfish-like shapes, "=" means they can appear simultaneously, purple color indicates the local distribution of PNIPAm. **e** Effect of the membrane flexibility on the insertion of PNIPAm. A higher membrane flexibility results in a lower percentage of insertion which is reflected in a decrease of the Δa (reduced monolayer area difference).

overall shape is typically not spherical. To further verify the importance of hydrophobicity, negative control experiments were carried out where either PEG$_{2k}$ or PNIPAm was added to a solution of polymersomes at 0 °C (Supplementary Fig. 3). In both cases, when organic solvent was added, no protrusions were formed. Since the temperature is below LCST, PNIPAm stayed hydrophilic. However, after heating both samples to 21 °C (above the LCST of PNIPAm), protrusions on the membrane were formed in the PNIPAm addition experiment, while in the experiment where PEG$_{2k}$ was added to the polymersome solution no formation of protrusions was observed. This is expected as PEG$_{2k}$ does not have the ability to initiate an amphipathic transition.. These results suggest that PNIPAm specifically makes the mimicking of curvature formation in polymersomes feasible.

Interestingly, when more PNIPAm (250 µg) was added in the system, new nonaxisymmetric shapes were observed instead of polymersomes with tentacles. Cigar-like, rackets-like, boomerang-like and 3-armed starfish-like shapes were observed under TEM, SEM, cryo SEM and cryo TEM (Fig. 3 and Supplementary Fig. 4). Tubular structures instead of flattened structures were observed from the cryo-SEM (Fig. 3a3-c3). Similar shapes were observed with 500 µg and 1000 µg PNIPAm (Fig. 3b, c). When comparing with liposomes, similar axisymmetric shapes have also been predicted in simulation studies with a bilayer-couple model or an area difference elasticity model applied for the calculation of possible shapes, in which Δa was used to present the membrane area change[29,33,34]. According to the phase diagram[16], cigar-like,

rackets-like, boomerang-like and 3-armed starfish-like shapes are in a higher Δa region, which means the membrane composition changed more in those species when compared to the other shapes. In other words, more PNIPAm has been inserted into these polymersome membranes. Moreover, these shapes appear in a wide range of Δa simultaneously[16], resulting in mixed shapes in our samples (Fig. 3).

**Visualization of PNIPAm**. As revealed above, PNIPAm was presumed to insert into the membrane to promote curvature formation. Visualization of the local distribution of PNIPAm is essential to prove insertion. As PNIPAm is not clearly visible under TEM, a heavy metal staining technique must be applied to enhance the image contrast before examination, where metal ions will preferentially react with or localize themselves in polymeric material containing specific groups like aromatics, ethers, alcohols, amides and so on[35]. In order to view PNIPAm specifically, the selectivity of the staining agent must be taken into account. As polymersome membrane also contains PS chains with aromatic moieties and PEG with ether moieties, traditional staining agents like ruthenium tetraoxide ($RuO_4$) and osmium tetroxide ($OsO_4$) will have an effect on PS and PEG as well. Heavy metal ions like $Co^{2+}$, $Ni^{2+}$, $Cu^{2+}$, $Zn^{2+}$, $Cd^{2+}$, and $Hg^{2+}$ play important roles in the formation of peptide metal ion complexes based on coordination between metal ions and amides, which inspired us to use of $Cu^{2+}$ to coordinate to the PNIPAm amide[35,36]. In principle, sulfur and oxygen can act as donor atoms in the coordination

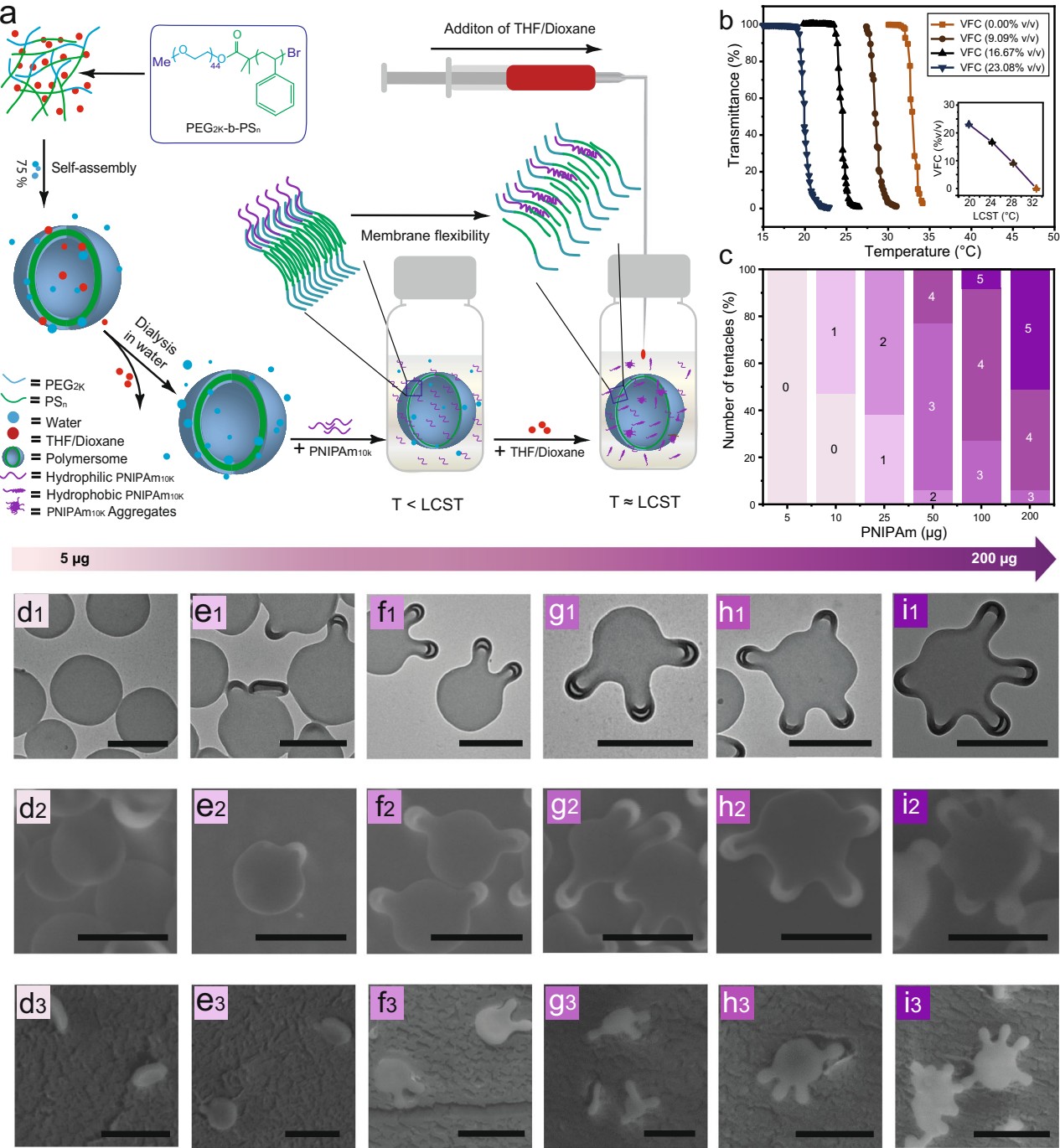

**Fig. 2 Curvature formation of polymersomes via PNIPAm insertion. a** Schematic representation of the curvature formation of polymersomes. The variation of LCST of PNIPAm in the volume fraction of water-cononsolvent (THF:Dioxane = 4:1) detected by UV–vis spectroscopy (**b**). The relation between increase of formed tentacles and the PNIPAm concentration. 100 particles were analyzed from each sample (**c**). TEM(d1-i1), SEM (**d**2-**i**2), Cryo-SEM (**d**3-**i**3) images depicting the formation of the polymersome membrane curvature during an increase of the PNIPAm concentration (5 µg, 10 µg, 25 µg, 50 µg, 100 µg and 200 µg). Samples were quenched after addition of 23.08% organic solvent. Scale bar 500 nm.

with $Cu^{2+}$, however, a coordination geometry study reveals that the binding of metals occurs almost exclusively via side chains of the amino acid residues[37]. This gives PNIPAm the specificity in the staining procedure as there is no such side chain present in PEG. Different shapes with PNIPAm staining were examined after removal of the excess free PNIPAm in solution (Fig. 4 and Supplementary Fig. 5). Because of the entangled structure, one $Cu^{2+}$ would coordinate with 4 or 6 amides allowing for the visualization of PNIPAm (presented as black dots) via multiple

coordination, since one $PNIPAm_{10k}$ chain contains in principle 88 amides. Moreover, PNIPAm only appear in the membrane where the protrusions were generated at lower PNIPAm concentration (Fig. 4b, c and Supplementary Fig. 5a, b). In polymersomes with two tentacles, the PNIPAm black dots are not as intense, and no dots were observed in areas with no protrusion (Fig. 4b1-b2). In polymersomes with four tentacles, the PNIPAm black dots are darker than polymersomes with two tentacles with some dots observed in areas with no protrusion (Fig. 4c1-c2).

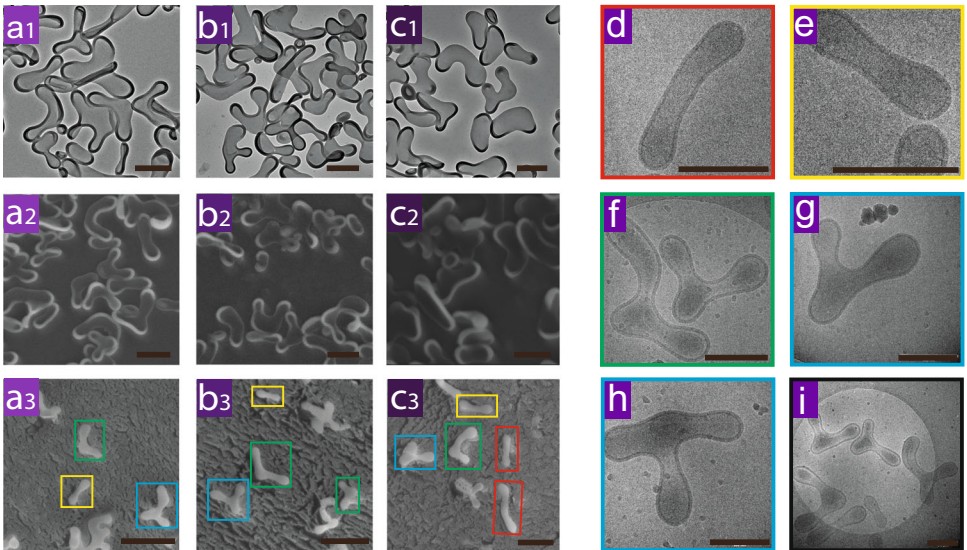

**Fig. 3 Polymersome membrane curvature changing with the increase of PNIPAm concentration. a** 250 µg, 500 µg (**b**) and 1000 µg (**c**)), examined by TEM(**b**1-**d**1), SEM (**b**2-**d**2), Cryo-SEM (**b**3-**d**3) and Cryo-TEM (**d**–**i**), in which nonaxisymmetric polymersomes with cigar-like (red box), rackets-like (yellow box), boomerang-like (green box), and 3-armed starfish-like (blue box) shapes were observed. Scale bar 500 nm.

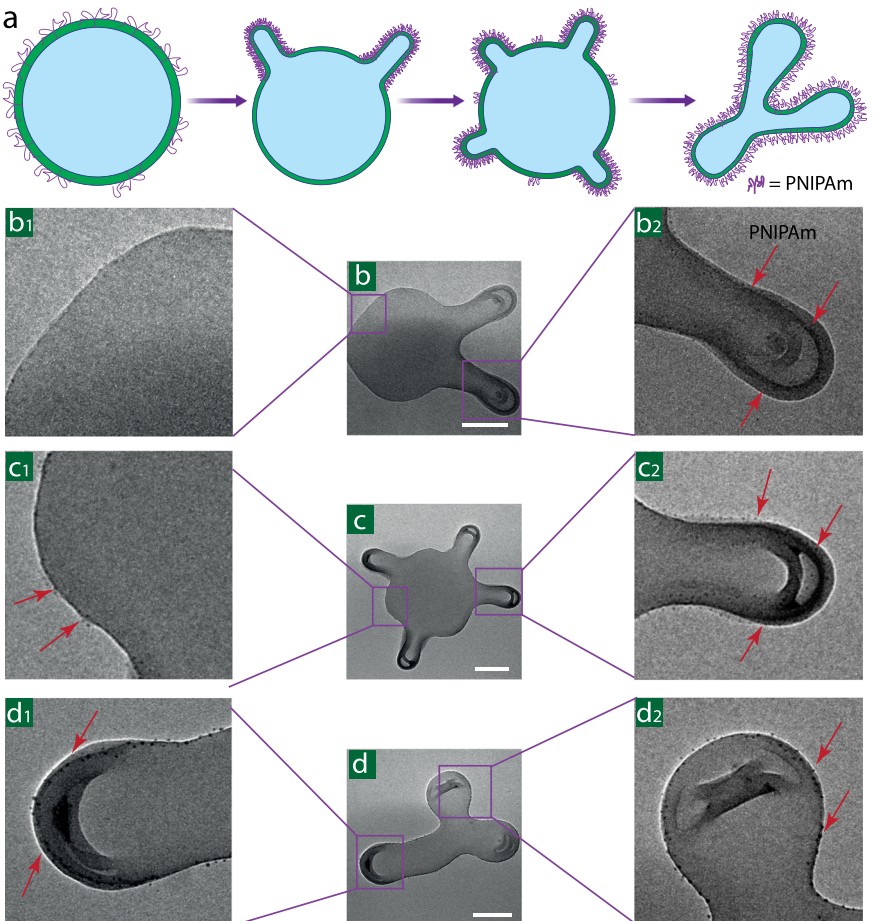

**Fig. 4 Visualization of PNIPAm by negative staining.** Scheme of PNIPAm distribution in the polymersome membrane (**a**). TEM image of polymersome membranes dyed with copper sulfate ($CuSO_4$). Coordination of $Cu^{2+}$ on PNIPAm makes the PNIPAm visible. Red arrows point towards the PNIPAm, visible as black dots (**b**–**d**). A zoomed-in area of the control area without PNIPAm (**b**1), with several dots of PNIPAm (**c**1) and PNIPAm aggregates (**d**1, **b**2-**d**2). Scale bar 200 nm.

When higher concentrations of PNIPAm were added (200 μg) to the polymersome solution and 5 tentacles were formed, PNIPAm black dots were observed almost all over the membrane (Supplementary Fig. 5c). Moreover, on the membrane of 3-armed starfish-like shape with the highest PNIPAm concentration (≥250 μg) added in the system, PNIPAm dots were present all over the membrane area with intense black color (Fig. 4d1-d2). A similar staining can also be found in cigar-like polymersomes (Supplementary Fig. 5d).

These results suggest that PNIPAm was inserted into all membrane areas when PNIPAm concentration reaches a certain threshold, accompanied by a shape transformation. In addition, oblate polymersomes without PNIPAm were also stained as a negative control, shown in Supplementary Fig. 6, where no specific black dots were detected. This observation is consistent with our previous hypothesis. With higher PNIPAm concentration, more PNIPAm can be inserted into the membrane, which leads to an increase of Δa and a simultaneous formation of variable polymersome shapes. To further prove the insertion of PNIPAm, Nuclear Magnetic Resonance spectroscopy (NMR) was employed. The presence of PNIPAm is demonstrated both in polymersome solution (Supplementary Figs. 7 and 8) and after dissolving the samples in CDCl₃ (Supplementary Fig. 9). We also quantified the amount of PNIPAm, which was determined to be 0.16 mg in the membrane of a 10 mg sample of PEG-*b*-PS. After ensuring all free PNIPAm was washed out, we found that the peaks from the inserted PNIPAm are significantly downshifted, broadened and less intense compared to the peaks of the free PNIPAm. The downward shift of the peaks could be due to increased hydrophobicity induced by the packing of PNIPAm and the PEG shielding. The peaks are broad and small because most inserted PNIPAm is in the solid phase, which cannot be seen on liquid NMR. Only small parts of the PNIPAm chain on the surface are slightly more dissolved by the solvent, making them liquid-like, resulting in small and broad peaks[38,39]. Since PNIPAm is inserted in the polymersome membrane and measured in solution (D₂O), broad peaks were observed from the backbone as it is in a solid phase and is covered by the PEG layer. More evidence is also provided by diffusion NMR. Free PNIPAm and polymersomes with free PNIPAm added in the solution showed a similar diffusion coefficient (Supplementary Figs. 10 and 11). Polymersome sample with PNIPAm inserted in the membrane showed almost no decay of the peak. Similar results were observed from the PEG peak of the polymersome membrane, as polymersomes are too big to diffuse within the detection limit of the probe, and anything that is attached to the polymersome will diffuse with a similar speed during the process (Supplementary Fig. S12). This indicates that PNIPAm is inserted in the polymersome membrane. The above results provide us a direct evidence for the insertion of PNIPAm into the polymersome membrane, which provides us an experimental platform to study the protein membrane interaction, previously only investigated by molecular modeling.

**The segregation of PNIPAm in polymersome membrane**. To investigate the details of curvature formation, samples were taken at different time points after the addition of plasticizing solvent. As shown in Fig. 5, the tentacles observed previously did not form immediately. Interestingly, triangular oblate shapes appeared in the presence of 10 μg PNIPAm and polygonal oblate shapes appeared in the presence of 100 μg PNIPAm at 23.08% VFC (Fig. 5b, f), meaning that the insertion of PNIPAm occurs in a random fashion without selectivity. However, after 10 min, the shapes change to polymersomes with one tentacle and 3-4 tentacles respectively (Fig. 5c–d, g–h). This rearrangement of

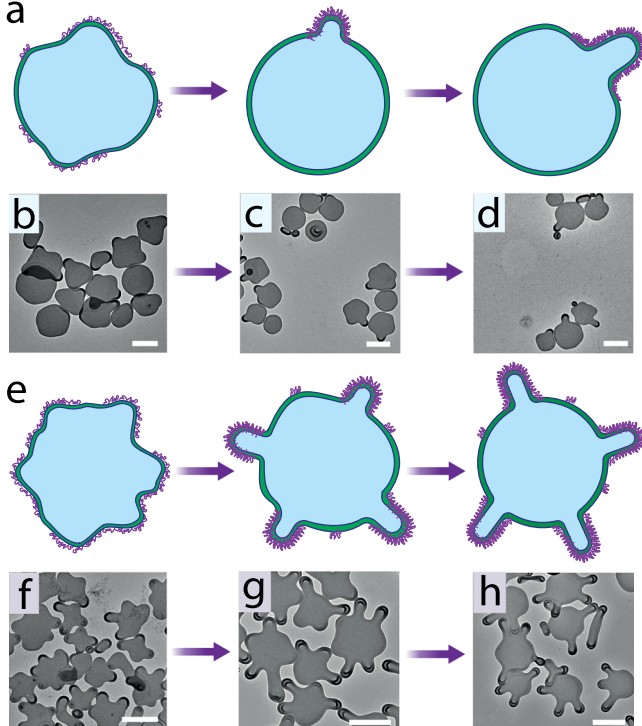

**Fig. 5 The segregation of PNIPAm in polymersome membrane.** Schematic representation of molecular rearrangement with the addition of PNIPAm 10 μg (**a**) and 100 μg (**e**) at 23.08% VFC. Tracking of membrane rearrangement by TEM at different time points (0 min (**b**, **f**), 10 min (**c**, **g**), 20 min (**d**, **h**) with the addition of PNIPAm 10 μg (**b–d**) and 100 μg (**f–h**). Scale bar 500 nm.

PNIPAm has been followed with the copper ion negative staining method. As shown in Supplementary Fig. 13, PNIPAm was first uniformly distributed in the membrane at 0 min, and then enriched in the protrusions of the membrane during a time period of 20 min. Previous studies have shown that the local composition changes in phospholipid bilayers can be induced by the change of the external environment, such as temperature, pH and osmolarity based on the high membrane fluidity[40,41], which results in phase separation in the liposome membrane[42–44]. In contrast, the polymersome membrane has low fluidity because of the higher molecular weight of the block copolymers and chain-chain entanglements during the assembly[45]. Here, the amount of added plasticizer is not enough to reassemble the PEG-*b*-PS membrane, due to the strong chain-chain entanglements[46]. Instead, the single polymer chain PNIPAm can locally diffuse along the membrane[47]. Firstly, the polymer length is much shorter than PEG-*b*-PS; secondly, the PNIPAm is not totally inserted into the entangled PS but partially embedded in between PEG, which is demonstrated by its detection via solution NMR. As shown in Supplementary Fig. 8, when polymersomes with PNIPAm inserted were dispersed in D₂O, PNIPAm could be detected by ¹H-NMR, but not PS. This suggests that PNIPAm is not entangled in the PS segments and therefore maintains its ability to diffuse. We assume the tentacle formation is mainly caused by the segregation of PNIPAm facilitated by the mobility of polymer chains along the membrane. The segregation of mixed polymer nanoscale vesicles has been considered to be difficult to achieve[48,49], however, as an post-inserted polymer, PNIPAm has more freedom than PEG-*b*-PS, which allows it to segregate within the membrane. On the other hand, a lateral phase separation of the polymer is expected in case of mixed polymers, due to its low

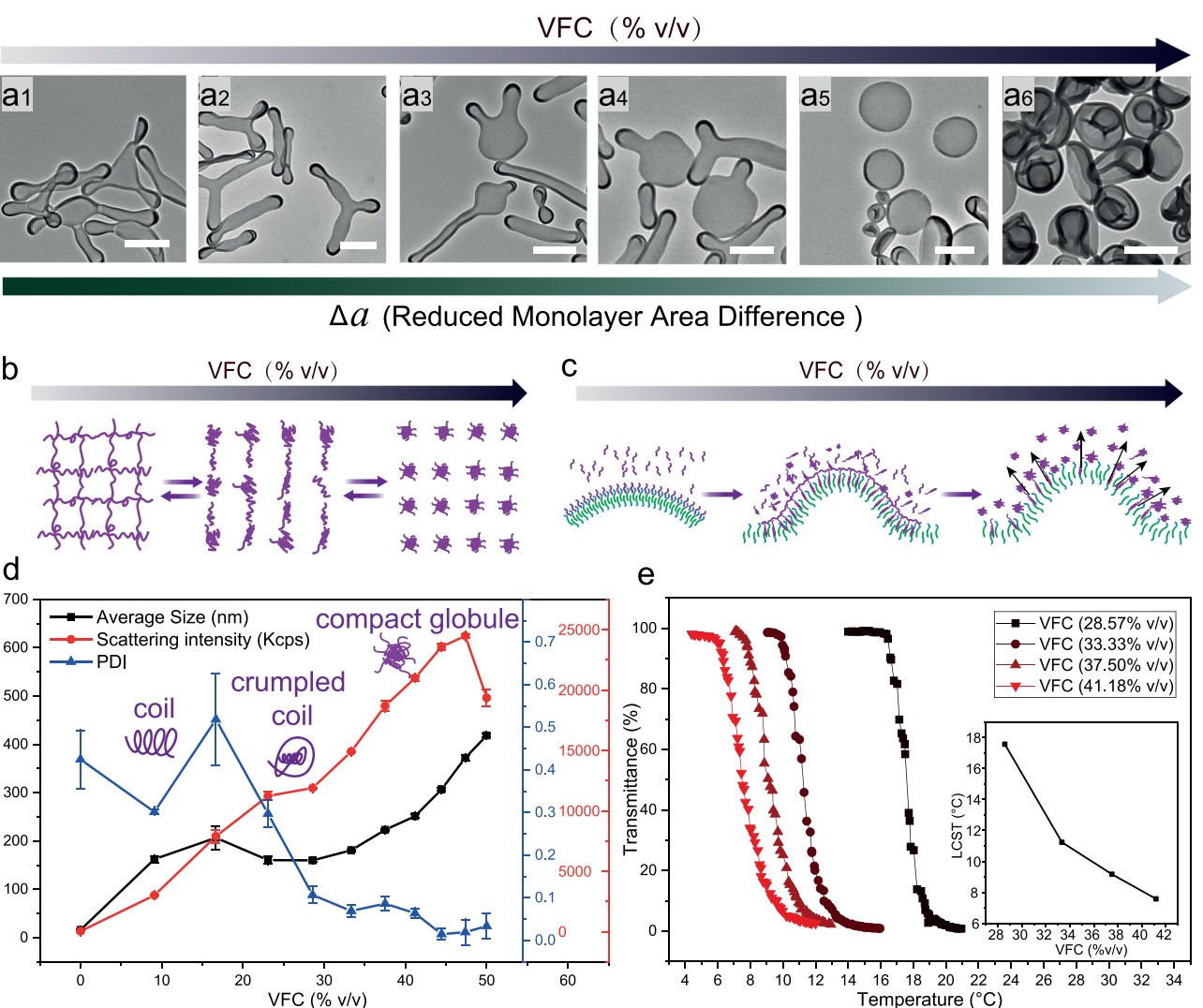

**Fig. 6 Reversible PNIPAm-membrane Interaction.** TEM image of polymersomes depicting changes in the membrane curvature during the increase of VFC ratio from 23.07% to 37.5% (black arrow). The reduced monolayer area difference decreases by increasing VFC ratio (green arrow) (**a**1–**a**6). Schematic representation of PNIPAm aggregation (**b**) and PNIPAm membrane interaction (**c**) with the increase of VFC ratio. Dynamic light scattering measurements for the observation of the aggregation of PNIPAm with the change of VFC, the structure of PNIPAm changed from coil, crumpled coil to compact globule. Values are mean ± SD (**d**). PNIPAm transmittance measurement by UV–vis spectroscopy to detect the shift of LCST with the change of VFC (**e**). Scale bar 500 nm.

mixing entropy. Similar segregation behavior was also observed in liposomes by M. Kühner and co-workers[44]. Moreover, during the segregation, polymersomes' membrane became thicker at the protrusions and thinner everywhere else, keeping the structure stable since the thick-walled compartments exhibit larger spontaneous curvatures than the thin-walled compartment.

**Reversible PNIPAm-membrane interaction.** The curvature formation in natural membranes is reversible, meaning that proteins can be dissociated and reused in a new cycle of vesicle formation[50]. To investigate the reversibility of our model, more organic solvent was added to the system in presence of 250 µg PNIPAm. 3-armed starfish-like shapes were observed with Δa around 1.6 at 23.07% VFC, and rackets-like shape and cigar-like shape (Δa ~ 1.4/1.5)[16] appeared at 25.39% VFC (Fig. 6a1-a2), followed by the appearance of two tentacles polymersomes with 27.63% VFC (Fig. 6a3) and one tentacle polymersomes (Δa ~ 1.2)[16] with 31.79% of VFC (Fig. 6a4). When the VFC percentage reached 33.79%, all polymersomes have transformed into oblate structures (Δa ~ 1.0) (Fig. 6a5), and into stomatocytes (Δa ~ 1.0) at 37.5% of solvent

(Fig. 6a6). With the increase of the VFC, the shape changing ends up with stomatocytes, which is similar to the shape transformation without the addition of PNIPAm (Supplementary Fig. 14), meaning that almost no membrane composition difference between the two monolayers is present. This might suggest that the PNIPAm slowly dissociates from the membrane while the membrane becomes more and more flexible. The reversibility of the curvature can be achieved by increasing the membrane flexibility. Moreover, this PNIPAm dissociation was also detected in other shapes with different PNIPAm concentrations (Supplementary Fig. 15). The assemblies with various shapes in the presence of different concentrations of PNIPAm all altered to stomatocytes shape at 37.5% VFC. To examine the dissociation of PNIPAm we used fluorescence resonance energy transfer (FRET) technique. PS linked with Cyanine3 (Cy3) was embedded in polymersomes during the assembly, while PNIPAm linked with Cyanine5 (Cy5) was used to interact with the self-assembled polymersomes. The interaction between PNIPAm and PS segments can be demonstrated by following the fluorescence intensity change from Cy5 while exciting Cy3. As shown in Supplementary Fig. 16, Cy5

fluorescence emission spectra (640–700 nm) were obtained by excitation of Cy3 at 512 nm. After addition of 150 μl plasticizer to the system, the PNIPAm-Cy5 inserted into the polymersome membrane. When Cy3 (embedded in the polymersome membrane during self-assembly) was excited at 512 nm, Cy5 was also excited by the emission from cy3 (blue line), causing it to emit light. After addition of 300 μl plasticizer to the system, the PNIPAm-Cy5 slowly dissociated from the polymersome membrane. When Cy3 was excited at 512 nm, Cy5 was slightly excited by the emission from Cy3, but to a much lesser extent than in the case when 150 μl plasticizer was added (green line). Furthermore, a Cy3 fluorescence quencher-Black Hole Quencher 2 (BHQ2) linked to PNIPAm was also applied to follow the dissociation. Similar results were observed as shown in Supplementary Fig. 17. The fluorescence from Cy3 was quenched when PNIPAm-BHQ2 was inserted into the membrane at 150 μl organic solvent addition (red line). However, after 300 μl plasticizer was added to the system, Cy3 fluorescence restored to normal (blue line), suggesting the dissociation of PNIPAm at higher VFC. We reasoned that the morphology of PNIPAm itself might have changed with the increase of VFC. On the other hand, the more and more flexible membrane gives PNIPAm more freedom to diffuse out of the membrane, as it is not anchored to the polymers. To prove this hypothesis, the behavior of PNIPAm under the same condition was detected using dynamic light scattering (DLS) technique. The scattering intensity of PNIPAm samples kept on rising with the increase of VFC, indicating that more and more PNIPAm particles with larger sizes were formed (Fig. 6d and Supplementary Fig. 18). The PNIPAm particle formation is based on the variation of LCST (Fig. 6e and Supplementary Table 1), the increase of VFC from 28.57% to 41.18% led to the shift of LCST from 17.6 °C to 8 °C, meaning that more and more aggregation was generated as PNIPAm was largely dehydrated. Moreover, the particle size was also stabilized during addition, except for a fluctuation at around 16.67%, caused by the process of PNIPAm aggregation. Since the LCST of PNIPAm with 16.67% VFC is 24.4 °C (Supplementary Table 1), which lies under the measurement temperature (25 °C), PNIPAm was slowly becoming hydrophobic, which explained the highest polydispersity index (PDI) with a value around 0.5 and several peaks in the size distribution (Supplementary Fig. 18a). A single peak appeared in the size distribution when 23.07% of organic solvent was added (LCST = 19.9 °C), meaning the formation of aggregated PNIPAm particles were slowly stabilized (Supplementary Fig. 18b). DLS was used for the study of coil to globule transition of individual PNIPAm chains with an extremely dilute PNIPAm solution (~5 μg/mL). The size of the PNIPAm chains largely decreased because of the collapse of the chains[51,52]. In our experiment, a concentration (0.1 mg/mL) similar to that of the PNIPAm present in our polymersome system was applied, which is much higher than in the PNIPAm chain solution, suggesting that interchains aggregation is the main reason for the particle formation. The larger size of PNIPAm particles, in the range of hundreds of nm in our experiment provides additional evidence. According to the above investigation we assume that the PNIPAm inserted into the polymersome membrane in a slightly entangled but not compacted state (Fig. 6b). With the increase of VFC, the inserted PNIPAm diffuses out because of the aggregation of the PNIPAm chains, so that less and less PNIPAm is staying in the membrane, resulting in a lower and lower Δa (Fig. 6c). Based on the results, we revealed that the curvature formation of polymersomes is reversible when membrane flexibility increases and PNIPAm is expelled from the membrane, higher flexibility might lead to a lower insertion ratio.

To conclude, we described a new approach to mimic biomembrane curvature formation using a polymersome model. By applying PNIPAm with adjustable amphipathicity to interact with polymersome membrane, different types of curvatures were

revealed which is evident from the polymersome-based nonaxisymmetric shapes formed in the process. The inserted PNIPAm changes the composition of the outer layer of the polymersome bilayer and results in a shift of Δa, thus, providing the prerequisite for curvature formation. The segregation of PNIPAm along the membrane then completes the curvature formation. These results provide evidence for the feasibility of mimicking curvature formation in polymersome models. In addition, insertion ratio in the membrane can be fine-tuned by PNIPAm concentration. High concentration of PNIPAm can lead to a higher membrane insertion ratio on polymersome membranes. Furthermore, higher flexibility might result in lower insertion because of the dissociation of PNIPAm. Due to the similarities in structural features, this study can provide experimental evidence for a further understanding of biomembrane curvature formation. This new approach of shape changing can also promote the development of new nonaxisymmetric polymersome-based nano-systems for various applications. In the future, this new strategy can be applied for the investigation of fundamental cellular processes like endo- and exocytosis, division and immune response.

## Methods

**Materials**. All reagents and chemicals were purchased from commercial sources and used as received. MilliQ-water (18.2 MΩ) was used in all the experiments. Poly (N-isopropylacrylamide), carboxylic acid terminated with average Mn 10,000 was purchased from Sigma-Aldrich. Transmission electron microscopy (TEM) samples were prepared in the following way: a solution of sample (5 μL) was air-dried on a carbon-coated Cu TEM grid (200 mesh). JEOL TEM 1400 microscope at an acceleration voltage of 120 kV and JEOL TEM 2100 at an acceleration voltage of 200 kV were used for the characterization of shape transformation. (Cryo-)SEM was performed on a JEOL 6,330 Cryo Field Emission Scanning Electron Microscope at an acceleration voltage of 3 kV in cryo-mode and 10 kV in dry mode.

**Preparation of polymersomes**. PEG$_{44}$-b-PS$_{172}$ (10 mg) was dissolved in 1 mL organic solvent mixture of tetrahydrofuran (THF) and 1,4-dioxane at the ratio of 4:1 by volume with a magnetic stirring bar. After dissolving the solution for 30 min, 3 mL of Milli-Q water was added to the solution at a rate of 1 mL h$^{-1}$ at room temperature under vigorous stirring (900 rpm) upon which polymersomes are formed. The suspension was then transferred to a dialysis membrane (SpectraPor, molecular weight cut-off: 12,000–14,000 and dialyzed against pure water (1000 mL) for 48 h with a frequent change of water.

**Solvent addition method for PNIPAm insertion into the polymersome membrane**. Firstly, 490 μL rigid polymersomes solution with a polymer concentration of 2 mg/ml was transferred into a 5 mL vial. 10 μL of PNIPAm with various concentrations (with the amount of PNIPAm = 5 μg, 10 μg, 25 μg, 50 μg, 100 μg, 200 μg, 250 μg, 500 μg, 1000 μg) was then added into the polymersomes solutions respectively. After ten minutes of mixing, THF: dioxane (4:1 v/v) mixture was added via syringe pump under a stirring speed of 900 rpm by using a stirring plate with a rate of 300 μL h$^{-1}$. Samples were withdrawn and quenched as scheduled. TEM, SEM, Cryo TEM and Cryo SEM samples were prepared to follow the membrane curvature formation.

**PNIPAm rearrangement over time**. 490 μL rigid polymersomes solution was transferred into a 5 mL vial followed by addition of 10 μg or 100 μg of PNIPAm and organic solvent addition as described above. Samples were withdrawn after solvent addition and rearranged for 0 min, 10 min and 20 min before quenching for TEM examination.

**Detection of dynamic PNIPAm-membrane Interaction**. 490 μL rigid polymersomes solution was transferred into a 5 mL vial. 250 μg of PNIPAm was then added into the polymersome solutions. After ten minutes of mixing, THF: dioxane (4:1 v/v) mixture was added via syringe pump under a stirring speed of 900 rpm by using a stirring plate with a rate of 300 μL h$^{-1}$. Samples were taken at different time intervals (30 min, 45 min, 40 min, 45 min, 50 min, 60 min), quenched with MilliQ water and examined via TEM. Similar procedure was applied to stomatocyte polymersomes, where samples were taken at 30 min, 45 min, and 60 min, quenched and examined with TEM.

**Size examination of PNIPAm aggregates formation**. Dynamic light scattering (DLS) experiments were studied on a Malvern Zetasizer Nano S equipped with a He-Ne (633 nm, 4 mW) laser and an Avalanche photodiode detector at an angle of

173° at 25 °C. Different concentrations of PNIPAm MilliQ water solutions were filtered through a 0.22 μm filter to remove the possible dust. Pure THF and 1,4-dioxane with a ratio of 4 :1 was added dropwise before measurement. The final concentration of PNIPAM in different water/organic solvent mixtures was kept at 0.1 mg/mL.

**The transmittance measurements of PNIPAm.** PNIPAm samples were conducted at 500 nm on a JASCO V-630 UV–Vis spectrophotometer equipped with a thermoregulator (±0.1 °C) with deionized water as a reference (100% transmittance). The final concentration of PNIPAM in different water/organic solvent mixtures was kept at 2 mg/mL. The LCST values were determined at 50%.

**Negative staining of PNIPAm in polymersomes.** To avoid the influence of free PNIPAm in the solution, samples were washed 5 times by centrifuging down the particles and tossing the upper solution with free PNIPAm. 3 μL of the polymersomes solution was then added on top of the carbon-coated 400-mesh copper grid for 5 min absorption. The excess fluid was then blotted away with filter paper. 3 μL of CuSO₄ (250 mM) solution was added on top of the grid for staining. After 5 min staining, the excess fluid was then blotted away with filter paper. Samples were then dried for 1 day before examination.

## Data availability

All data generated and analyzed during this study are included in this article and its Supplementary Information, and also available from the authors upon reasonable request.

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

## Acknowledgements

D.A.W. acknowledges the NWO Chemiche Wetenschappen VIDI Grant 723.015.001 for financial support. We acknowledge support from the Ministry of Education, Culture and Science (Gravity Program 024.001.035). The authors would like to thank G.J.J. from General Instruments at Radboud University for assistance with the microscopy analysis.

## Author contributions

J.S. and D.A.W. designed the experiments. J.S., S.J.R., S.Z., J.L. performed the experiments, J.S. analyzed the results. J.S. and D.A.W. wrote the manuscript.

## Competing interests

The authors declare no competing interests.
