## [Peer Review File · Nature Communications]

REVIEWER COMMENTS

Reviewer #1 (Remarks to the Author):

This manuscript reports the mimicking of membrane curvature by inserting polymer into polymersome. Such "hollow" structural models are often used to study biomimetic. The experimental part is technically sound and is in general well supported with the conclusions. Despite the interesting topic, the novelty of the work is not at the sophisticated level required for the publication in Nature Communications. To justify this statement: none of the materials used in this work is new (neither PEG-b-PS nor PNIPAm); the idea of using polymersomes in biomimetics is well known (either as a biomimetic membrane, as a carrier, nanoreactors etc.). Taking advantage of the thermo-responsive behavior (amphiphilic nature) of PNIPAm in combination with other systems (biological and polymeric) is also being well investigated. The added-value of the manuscript is based on a "new" strategy that may be used in the future for studying various cellular processes. How could this approach be implemented or applied to hybrid systems, e.g. polymer-proteins?

Reviewer #2 (Remarks to the Author):

The paper presents novel results on the generation of curvature in PEG-PS polymersomes after PNIPAm helix insertion. The authors are able to visualize the insertion and quantify it with respect to PNIPAm concentration, solvent volume and temperature. The method can be used to generate non-axisymmetric polymeric vesicles for nanotechnological applications, although the authors do not provide specific examples of such needs and applications.

While the experimental approaches in the paper are novel, the results and conclusions are presented with significant degree of ambiguity. Below are my remarks in the order as they appear in the manuscript. I believe the text requires significant revision before publication.

1. The authors claim that the polymersomes system can provide insights on the curvature generation in biomembranes. However it must be noted that polymeric membranes have notably different mechanical properties, i.e. they are not fluid and are significantly more flexible. Therefore, even though the generation of curvature upon PNIPAm insertion appears similar to the insertion of amphipathic helices in lipid membranes, the mechanisms by which polymeric vesicles change shape may differ significantly from that in lipid vesicles. Further discussion is needed on the coupling between surface area and vesicle volume in polymersomes; on the formation of wrinkles observed in Figure 4; and on the redistribution of membrane components in non-fluid polymeric membranes.
2. Membrane flexibility and mobility are not necessarily the same thing. The first refers to the elastic properties of the membrane, whereas the second could describe its fluid nature. However on page 5 they are used interchangeably (line 101 and 114). Can the authors please clarify the sentence on lines 113-114. What is the equilibrium point? Equilibrium in terms of what?
3. What do the authors mean by "the edge of membrane" on page 6, 139? Edge is usually used in conjunction with pores. Do they mean "tip of the protrusions"?

4. Page 8, line 174- Which phase diagram?

5. What is the reduced volume of the vesicles on Figure 3, i.e. does it stay constant as the PNIPAm concentration increases? Vesicle shapes depend on both Δa and the reduced volume, as in e.g. (H. Doebereiner, Current Opinion in Colloid & Interface Science 5 -2000. 256-263)

6. What is the dynamics, i.e. time dependence of PNIPAm insertion in the membrane? Does it depend on the degree of membrane deformation? It is known that curved membranes facilitate insertion. Is it possible that the segregation shown in Figure 5 is not due to lateral rearrangement of PNIPAm, but due to facilitated adsorption at sites of high curvature? Related to this, do the authors have any proof of the statement on page 14, lines 274-275? Furthermore, I do not understand the sentence " Moreover, ..." lines 275-278.

7. How do the authors quantify Δa , page 14, lines 286-292?

8. How can membrane flexibility facilitate PNIPAm desorption, line 297? How is PNIPAm anchored to the polymers (line 304)?

9. What exactly are the conditions referred to in line 305- PNIPAm in VFC solution only or also in the presence of vesicles? How are the molecular arrangement and transitions on Fig. 6b deduced?

10. The sentence on lines 337-340 requires revision.

(M.Staykova)

Reviewer #3 (Remarks to the Author):

This interesting contribution by Wilson and coworkers shows the shape transformation of polymer vesicles by introducing hydrophobic macromolecular guests to the block copolymer (BCP) bilayer. They used the LCST behavior of PNIPAM in the presence of organic solvents in water, which lowers the LCST temperature of the thermosensitive polymer proportional to the volume fraction of organic solvents in water. The authors had their motivation to mimic the behavior of biomembranes in the presence of membrane-bound proteins, which are responsible for the change of the curvature of lipid bilayers. The introduction of hydrophobic nano-sized objects such as nanoparticles to the block copolymer bilayer has been utilized to implement functional elements to polymer vesicles. However, it is scarce to use this idea for the intentional shape transformation of polymer vesicles. I am enthusiastic about this paper because the results reported herein could open a new way to manipulate the shape of polymer vesicles rationally by introducing membrane-protein mimics. If realized, this can be significant for understanding how the polymer membrane behaves under perturbation arising from guests. The expansion of the potential of polymer vesicles for nanoscience and drug delivery would be an excellent prospect as well. However, I believe that the presented ideas of this paper should be carefully understood and applied to explain the data presented in this

paper in terms of the fundamentals of the shape change of polymer membranes. Let me elaborate on my opinions.

Polymer vesicles consist of block copolymer bilayers, which are conceptually similar to lipid bilayers. BCP bilayers, however, are different from lipid bilayers in terms of physical robustness arising from the high molecular-weight of BCPs. This difference has to be considered when the analogy between BCP bilayers and lipid bilayers as the translational motion of a BCP within the bilayer can be extremely limited. Earlier studies by Eisenberg and others showed a very low critical micelle concentration of block copolymer micelles and limited (virtually non-existent) molecular exchange between polymer micelles. I will summarize how polymer vesicle changes their shape in response to external stimuli with these in mind. With a given internal volume and the surface area of the BCP bilayer, polymer vesicles adopt a spherical morphology in solution. Polymer vesicles change their shape when (1) the volume of the internal compartment changes; (2) the surface area of the BCP bilayer changes. As I pointed out, BCP bilayers lack fluidity because of the limited translational motion of BCPs in the bilayer. Some hydrophobic polymers such as polyisoprene and polyethylene, having low glass transition temperatures provide fluidity to the BCP bilayers if the molecular weight of these polymers is limited (Bates and coworkers). BCP bilayers built with the high glass transition temperature hydrophobic polymers such as polystyrene can be fluidized by plasticizing polystyrene with organic solvents. The shape transformation of polymer vesicles with plasticized membranes can change their shape from a sphere to a discoid and stomatocyte upon the decrease in the volume of the internal compartment caused by the osmotic ejection of organic solvents. Wilson and coworkers are the experts at the forefront in this field.

In this paper, organic solvents (THF/dioxane) were introduced to the aqueous dispersion of spherical polymer vesicles in the presence of solubilized PNIPAM. The solvent molecules plasticize polystyrene membrane of polymer vesicles and destabilize PNIPAM to be hydrophobic; (3) infiltrate to the membrane to increase the internal compartment volume. Figure 1 shows that the initial result of the shape transformation is discoid, resulting from the decrease of the volume decrease of the inner compartment. These EM images raise one question: is the presented process resulting from the increase or decrease of the internal volume?

The structures look flat in some of the EM images presented in this paper. The destabilized PNIPAM is collapsed to hydrophobic nanoparticles (presumably single-chain particles), which entered into the hydrophobic polystyrene part of the membrane. The authors claimed that this inclusion of nanoparticles perturbed the curvature of the membrane and led to the formation of the buds (tentacles) around the perimeter of the discoid. Considering the nature of BCP bilayers that involve the interdigitation of polymer chains (as described in Fig 1), the hydrophobic polymer chains most likely would form a highly entangled domain. When the hydrophobic nanoparticles of PNIPAM single chains entered into the BCP bilayer, this event might increase the volume fraction of the hydrophobic domain compared to the PEG domain. The increase of the hydrophobic domain's volume also might change the morphology of polymer vesicles. The morphological change would be lower curvature structures like flat lamella. My next question is: are the structures after PNIPAM introduction vesicles or flat structures?

It is claimed that incorporated PNIPAM chains were localized at the end-cap of the protrusion (tentacle). The authors showed that the small particle-like electron-rich dots were mostly found from the ends of the tentacles. However, I am not convinced that Cu^{2+} binds with the amides of

PNIPAM. Even in the case of Cu-binding to collapsed PNIPAM chains, I am not sure that this technique can provide enough contrast in a single molecule level, which allows the trace of the location of PNIPAM chains in the membrane. Can the authors have a more effective method to prove this?

The segregation of embedded PNIPAM chains within the polymersome membrane is highly unlikely when one considers the viscosity of the solvent-swollen polystyrene domain. This claim should be carefully made based on experimental facts, not solely based on TEM images.

When additional organic solvents were introduced to the polymersomes with embedded PNIPAM, the polymersomes underwent further shape changes from tubular structures to oblates and stomatocytes. This transition indicates that the internal volume of the polymersome must be reduced with the given surface area. Do more organic solvents induce osmotic swelling of the inner compartment by introducing more solvent molecules? Could the shape change happen upon quenching the self-assembly? The extraction of embedded PNIPAM with additional organic solvents also has to be proved by experiments.

Additional comments are

The Δa values shown on page 14-15 seem unsubstantiated. Please provide how these numbers can be used.

Why was only 10k PNIPAM used for the experiments? How about PNIPAM with other molecular weight?

Line 241, change Figure 10-11 to Figure S10-11.

REVIEWER COMMENTS

Reviewer #1 (Remarks to the Author):

This manuscript reports the mimicking of membrane curvature by inserting polymer into polymersome. Such "hollow" structural models are often used to study biomimetic. The experimental part is technically sound and is in general well supported with the conclusions. Despite the interesting topic, the novelty of the work is not at the sophisticated level required for the publication in Nature Communications. To justify this statement: none of the materials used in this work is new (neither PEG-b-PS nor PNIPAm); the idea of using polymersomes in biomimetics is well known (either as a biomimetic membrane, as a carrier, nanoreactors etc.). Taking advantage of the thermo-responsive behavior (amphiphilic nature) of PNIPAm in combination with other systems (biological and polymeric) is also being well investigated. The added-value of the manuscript is based on a "new" strategy that may be used in the future for studying various cellular processes. How could this approach be implemented or applied to hybrid systems, e.g. polymer-proteins?

Answer: We thank the reviewer for the comments. To mimic the structure of cell membrane, lipid-based liposomes and polymer-based polymersomes have been developed and used as carriers and reactors. However, cellular membranes are not just double-layered membrane, they can also respond to changes in the environment to implement the fundamental cellular processes, such as endo- and exo-cytosis, immune response and cell motion. The mechanisms behind these complex processes are essential for us to understand cells and life, which are not yet clear. To make use of these biomimetic vesicles to further explore and mimic the cellular behaviours will not only provide us valuable information about these cellular processes but also assist us to develop smart vesicles for nanoscience, drug delivery etc. During the cellular processes, biomembrane curvature formation has long been observed to be crucial to maintain the functionality and shape of organelle, which was studied in this manuscript via using polymersome as model and Poly(N-isopropylacrylamide) (PNIPAm) to induce the local curvature inspired by protein-membrane interaction. Indeed, the thermo-responsive behaviour of PNIPAm has been widely used in nanoparticle systems as part of the diblock polymers or hydrogels however, we mainly take advantage of its co-nonsolvency phenomenon, use this polymer as a substitute for amphipathic helices to investigate the interaction between hydrophobic blocks and bilayer membranes, which has not been studied before. The novelty of our study doesn't rely on development of new materials but in mimicking cellular processes, it was in our intention to use materials with well-known properties for testing our concept, in fact their known properties is an advantage for our study.

In addition, several polymersome-based nonaxisymmetric shapes has been observed for the first time while fine-tuning the amount of PNIPAm. To the best of our knowledge, this is the first-time polymer insertion has been achieved, protrusions being formed in polymersomes, and also the first-time detailed information on curvature generation has been revealed in membrane model. These results proved the capabilities of using polymersomes as biomembrane mimicking model in the level of membrane composition change for broadening the investigation of membrane insertion and curvature formation. Meanwhile, polymer insertion induced shape changing would open up routes for designing of non-axisymmetric nanocarriers, nanomachines etc to enrich the infinite possibilities of nanotechnology. For example, it is believed that the shape of nanocarriers can influence cellular uptake, however, researches about this topic has been restricted by the shapes that we can fabricate no matter from liposomes or polymersome or any other materials. With the method we have here, more non-axisymmetric nanocarriers can be fabricated, which will definitely help us to deeper the

understanding of this relationship. In terms of nanomachines, our group is trying to develop nanomotors with controlled motion, these non-axisymmetric nanocarriers will be good models for it. Moreover, the Brownian motion of spherical particles has been observed and described by Stokes-Einstein equation, however the motion of non-spherical and non-axisymmetric particles has not been studied and defined especially in nanoscale. We hope that our new findings can also help to gain the knowledge of this field. With the idea from this paper, the hybrid systems, like protein-polymer systems, can also benefit from it. For example, modifying proteins with polymers containing amphipathic helices property might help proteins to anchor on nanoparticles. In terms of in vivo protein- cell interaction, the modification of amphipathic helices polymers on proteins might increase the efficacy.

Our knowledge of this field suggests that the importance of this timely work will be appreciated by researchers from diverse backgrounds and hope that the reviewer will agree on the novelty and importance of this study, as the other reviewers have noted.

Reviewer #2 (Remarks to the Author):

The paper presents novel results on the generation of curvature in PEG-PS polymerosomes after PNIPAm helix insertion. The authors are able to visualize the insertion and quantify it with respect to PNIPAm concentration, solvent volume and temperature. The method can be used to generate non-axisymmetric polymeric vesicles for nanotechnological applications, although the authors do not provide specific examples of such needs and applications.

While the experimental approaches in the paper are novel, the results and conclusions are presented with significant degree of ambiguity. Below are my remarks in the order as they appear in the manuscript. I believe the text requires significant revision before publication.

1. The authors claim that the polymerosomes system can provide insights on the curvature generation in biomembranes. However, it must be noted that polymeric membranes have notably different mechanical properties, i.e. they are not fluid and are significantly more flexible. Therefore, even though the generation of curvature upon PNIPAm insertion appears similar to the insertion of amphipathic helices in lipid membranes, the mechanisms by which polymeric vesicles change shape may differ significantly from that in lipid vesicles. Further discussion is needed on the coupling between surface area and vesicle volume in polymerosomes; on the formation of wrinkles observed in Figure 4; and on the redistribution of membrane components in non-fluid polymeric membranes.

Answer: We thank the reviewer for the comments. Indeed, liposomes and polymersomes are different, for example, the membrane of polymersomes (5-50nm) is much thicker than liposomes (3-5nm). Although liposomes use the same materials (lipids) as cell membrane, the properties of the cell membrane are different than liposomes while thickness of cellular membrane is 8-10 nm (including the lipid bilayer, membrane proteins and small molecules)¹. Polymersomes might have advantage in mimicking cellular membrane as it can be adjusted to similar thickness by tuning the length of polymers. In contrast, liposomes turned to be more unstable since it is not as thick as the cell membrane. Furthermore, the existence of membrane proteins and small molecules like cholesterol on the cellular membranes might not only increase their thickness but also makes the property different from only lipids membranes, which makes them more stable and flexible than liposomes. In terms of the fluidity, polymersomes are less fluidic than both liposomes and cellular membranes however in our case we used plasticizer to swell the membrane and increase its fluidity². Furthermore, in this work we study the segregation of PNIPAm, which is a post-inserted polymer that only interacts with

the membrane in a dynamic way. This means that PNIPAm is not totally inserted in the membrane, and the segregation is possible. Further discussion about the redistribution of PNIPAm is added in the manuscript:

To investigate the details of curvature formation, samples were taken at different time points after the addition of plasticizing solvent. As shown in Figure 5, the tentacles observed previously did not form immediately. Interestingly, triangular oblate shapes appeared with 10 μg PNIPAm presented and polygonal oblate shapes appeared with 100 μg PNIPAm presented at 23.08% VFC (Figure 5b,f), meaning that the insertion of PNIPAm is random, with no selectivity. However, after 10 minutes arrangement, these shapes transitioned to one tentacle polymersomes and polymersomes with 3-4 tentacles (Figure 5c-d, g-h) respectively. This rearrangement of PNIPAm has been followed with the copper ion negative staining method. As shown in Figure S13, PNIPAm was first everywhere on the membrane at 0 min, and then moved to the protrusions of the membrane at 20 min.

Previous studies have shown that the local composition change of phospholipids bilayer can be induced by the change of external environment, such as temperature, pH and osmolality based on the high membrane fluidity^{3, 4}, which results in phase separation in liposome membrane⁵⁻⁷. In contrast, polymersome membrane has low fluidity because of the higher molecular weight of block copolymers and chain-chain entanglements during the assembly⁸. Here, the amount of plasticizer added is also not enough for the reassembly of PEG-PS, due to the strong chain-chain entanglements⁹. Instead, the inserted PNIPAm is more fluid than PEG-PS. Firstly, the polymer length is much shorter than PEG-PS; secondly, the PNIPAm is not totally inserted into the entangled PS but partially embedded in between PEG, which is demonstrated by its detection via solution NMR. As shown in Figure S8, when polymersomes with PNIPAm inserted were dispersed in D_2O , the PNIPAm could be detected by ^1H NMR, as well as PEG but not PS. This suggests that PNIPAm is not entangled in the PS segments, therefore it maintains its fluidity. We assume the tentacle formation is mainly caused by the segregation of PNIPAm, resulting in its mobility along the membrane. The segregation of mixed polymer nanoscale vesicles has been considered to be difficult to achieve^{10, 11}, however, as a post-inserted polymer, PNIPAm has more freedom than PEG-PS, which allows it to segregate within the membrane. On the other hand, a lateral phase separation of the polymer is expected when there are mixed polymers, due to its low entropy of mixing. Similar segregation behaviour was also observed in liposomes by M. Kihner and co-workers⁷. Moreover, during the segregation, polymersomes' membrane has become thinner at the normal place but thicker at the protrusions, which kept the structure stable since the thick-walled compartments exhibit smaller spontaneous curvatures than the thin-walled compartment.

Figure S8 ¹H NMR spectra (400 MHz, 298 K) in D₂O of polymersomes, with free PNIPAm added (top), free PNIPAm (middle) and polymersomes with inserted PNIPAm (bottom). A broad signal from the PEG is observed from 4.0 to 3.0 ppm, together with a sharp peak at 3.63, representing PEG at and sticking out of the surface of the polymersomes respectively. When PNIPAm is added in polymersomes without solvent addition, it cannot enter the membrane and the signal of free PNIPAm can be observed (top). When PNIPAm is added to polymersomes followed with solvent addition, it can enter the membrane, and presented as solid state, since broad signals can be observed at 1.2, 0.8 and 0.0 ppm. The change of microenvironment might cause the peak shift. Meanwhile, the signal of free PNIPAm is not observed since its washed over (bottom). The broadness of the peaks indicate insertion in the membrane.

Figure S13. Staining of PNIPAm distribution in polymersome membrane. TEM image of polymersome with the addition of 100 μg PNIPAm at 23.08% VFC, and membrane dyed with copper sulfate (CuSO_4) at different time point (a) 0 min and (b) 20 min. Scale bar 200 nm. PNIPAm was first everywhere on the membrane, and then moved to the protrusions on the membrane.

The reviewer also suggested to discuss the coupling between surface area and vesicle volume in polymersomes. We agree the surface area and vesicle volume is indeed very important for these structures, the calculation of surface area and vesicle volume for polymersomes has been discussed in our previous work¹², which is based on cylindrically symmetric polymersomes. In the current case, the protrusions formed from the membranes have made the calculations more difficult as most of the shapes we created are non-axisymmetric shapes. If we need to follow the experimental results, we have to actually cut the structure into pieces to know the exact parameters to calculate the volume, which is impractical as the structure is so small. At this moment, we are trying to find another calculation method, and hope to find a way to simulate the shape changes in this experiment. On the other hand, based on the simulation we have done before¹², we knew that the parameters in polymersomes' simulation is similar to liposomes in the same structures, which is why in this manuscript we compared the results we got with the liposome phase diagram.

In terms of the formation of wrinkles observed in Figure 4, this is actually due to the drying effect during the sample preparation, which can also be observed from the dry TEM pictures in Figure 2. If we look at the cyro-SEM (Figure 2 d) and cyro-TEM pictures (Figure S2), the wrinkles do not exist. As the protrusions formed on polymersomes are actually tubular-like structure instead of flat disk, they tend to be thicker after dried out comparing to the flat part of the structure.

2. Membrane flexibility and mobility are not necessarily the same thing. The first refers to the elastic properties of the membrane, whereas the second could describe its fluid nature. However, on page 5 they are used interchangeably (line 101 and 114). Can the authors please clarify the sentence on lines 113-114. What is the equilibrium point? Equilibrium in terms of what?

Answer: We thank the reviewer for the questions. We could have made it more clear about the flexibility and mobility after the addition of plasticizer. In fact, plasticizer can not only increase the flexibility but also untie the entanglement of the polymers in the membrane, which can result in the increase of the mobility in the membrane^{13, 14}. Here, the plasticizer is used to increase the space between the assembled polymers, so that PNIPAm can insert into the membrane. The equilibrium point here means the balance between the PNIPAm hydrophobicity and the possibility for PNIPAm to insert, mainly depends on the space between the assembled polymers. To make it clear we revised this part in the manuscript:

Interestingly, the addition of plasticizer can not only increase the flexibility, but also is stretching the membrane creating space between the assembled polymers for the PNIPAm insertion . When it reaches the point that PNIPAm becomes hydrophobic and the membrane has made some space for the insertion of PNIPAm, curvature formation was observed.

3. What do the authors mean by “the edge of membrane” on page 6, 139? Edge is usually used in conjunction with pores. Do they mean “tip of the protrusions”?

Answer: We thank the reviewer for the question. The edge of the membrane here meaning the rim of

the vesicular structure in Figure 2e3, as the PNIPAm was inserted there. To make it clear, we revised the manuscript:

Moreover, the subsequent curvature formation is also based on this flattening and widening vesicular structure (Figure 2e3-i3). Since the rim of this vesicular structure is more stretched and has the largest mean curvature compared to the rest of the membrane³³, more space between the assembled polymers is available between the entangled polymer chains for the insertion of PNIPAm.

4. Page 8, line 174- Which phase diagram?

Answer: We thank the reviewer for the question. Here we compared our results with the lipids simulation phase diagram from reference 16, as we are not yet able to simulate the polymersomes change, and it confirmed that polymersomes' parameters could be similar to the liposomes. We have added the reference to make it clear.

5. What is the reduced volume of the vesicles on Figure 3, i.e. does it stay constant as the PNIPAm concentration increases? Vesicle shapes depend on both Δa and the reduced volume, as in e.g. (H. Doebereiner, Current Opinion in Colloid & Interface Science 5 -2000. 256-263)

Answer: We thank the reviewer for the question. The reduced volume of the vesicles on Figure 3 is not known yet, as we haven't found a practical method to calculate it. We have to know the details of the shapes (the thickness of each part, the perimeters etc), as they are nonaxisymmetric shapes, we couldn't find a way to calculate it. However, we know that with this amount of solvent added, the shapes changed from spherical to disks (reduced volume around 0.6)¹², and the amount of PNIPAm can maybe give a bit of the extra osmotic pressure, it does not allow them to change to stomatocytes (reduced volume around 0.55). A control experiment with the same molar ratio PEG was added to prove this in Figure S3. Therefore, we believe that even when the reduced volume has been changed, the difference is marginal. Under this situation, the shape change should mostly depend on the Δa .

Figure S3. Temperature control in the insertion of PNIPAm. PEG 2k with no LCST transition was used for comparison. The transition of PNIPAm happened when increases the temperature from 0 °C to 21°C near LCST of PNIPAm. Sample with the addition of 100 µg PEG2k at 0 °C with 23.07% organic solvent (a) and increase the temperature to 21 °C (b). With the addition of 100 µg PNIPAm10k at 0 °C with 23.07% organic solvent (c) and increase the temperature to 21 °C (d).

6. What is the dynamics, i.e. time dependence of PNIPAm insertion in the membrane? Does it depend on the degree of membrane deformation? It is known that curved membranes facilitate insertion. Is it possible that the segregation shown in Figure 5 is not due to lateral rearrangement of PNIPAm, but due to facilitated adsorption at sites of high curvature? Related to this, do the authors have any proof of the statement on page 14, lines 274-275? Furthermore, I do not understand the sentence “Moreover, ...” lines 275-278.

Answer: We thank the reviewer for the comments. The dynamics mainly means that the insertion of PNIPAm can be controlled by dynamically adding organic solvent to the system. When membrane deformed to a flat disc structure, the edge is more stretched, which is the place for PNIPAm to insert. With the negative staining experiment, we have observed that the PNIPAm presented before the segregation. On the other hand, the high curvature can only be formed when PNIPAm is segregated and create a thicker membrane at the local curvature. Without the insertion, it should not happen as only osmotic pressure could not produce the high curvature. The statement on page 14 can be proved from the NMR detection in Figure S8. This part has been rewritten in the manuscript:

Previous studies have shown that the local composition change of phospholipids bilayer can be induced by the change of external environment, such as temperature, pH and osmolality based on the high membrane fluidity^{3, 4}, which results in phase separation in liposome membrane⁵⁻⁷. In contrast, polymersome membrane has low fluidity because of the higher molecular weight of block copolymers and chain-chain entanglements during the assembly⁸. Here, the amount of plasticizer added is also not enough for the reassembly of PEG-PS, due to the strong chain-chain entanglements⁹. Instead, the inserted PNIPAm is more fluid than PEG-PS. Firstly, the polymer length is much shorter than PEG-PS; secondly, the PNIPAm is not totally inserted into the entangled PS but partially embedded in between PEG, which is demonstrated by its detection via solution NMR. As shown in Figure S8, when polymersomes with PNIPAm inserted were dispersed in D₂O, the PNIPAm could be detected by ¹H NMR, as well as PEG but not PS. This suggests that PNIPAm is not entangled in the PS segments, therefore it maintains its fluidity. We assume the tentacle formation is mainly caused by the segregation of PNIPAm, resulting in its mobility along the membrane. The segregation of mixed polymer nanoscale vesicles has been considered to be difficult to achieve^{10, 11}, however, as a post-inserted polymer, PNIPAm has more freedom than PEG-PS, which allows it to segregate within the membrane. On the other hand, a lateral phase separation of the polymer is expected when there are mixed polymers, due to its low entropy of mixing. Similar segregation behaviour was also observed in liposomes by M. Kihner and co-workers⁷. Moreover, during the segregation, polymersomes' membrane has become thinner at the normal place but thicker at the protrusions, which kept the structure stable since the thick-walled compartments exhibit smaller spontaneous curvatures than the thin-walled compartment.

7. How do the authors quantified delta(a), page 14, lines 286-292?

Answer: We thank the reviewer for the comments. As we mentioned before, the calculation for delta(a) in polymersome is not yet possible at this moment, however, as the similarity between liposomes and

polymersomes has been proved, to make it easy to understand, we cited the delta(a) from liposomes in the explanation. To make it clear, we added the reference in the manuscript.

8. How can membrane flexibility facilitate PNIPAm desorption, line 297? How is PNIPAm anchored to the polymers (line 304)?

Answer: We thank the reviewer for the comments. As we mentioned before, the addition of solvent not only increased the flexibility of the membrane but also opened the space between the assembled polymers as the membrane is stretched¹⁵. Which allows the PNIPAm to insert due to the hydrophobic effect. However, with the increase of organic solvent, the space is getting bigger, which facilitates PNIPAm desorption. On the other hand, once PNIPAm is not so attached to the membrane, PNIPAm will be grabbed by the PNIPAm micelles formed in the solution, which assisted the desorption.

9. What exactly are the conditions referred to in line 305- PNIPAm in VFC solution only or also in the presence of vesicles? How are the molecular arrangement and transitions on Fig. 6b deduced?

Answer: We thank the reviewer for the comments. The conditions here refer to the same concentration of PNIPAm in solution with the absence of vesicles, and slowly add organic solvent to investigate the behaviour of PNIPAm. The illustration in Fig. 6b is based on the results from DLS in Fig. 6d, when PNIPAm dissolves in water solution, the polymers behave like coil structures, as its very small, does not have light scattered from them (Scattering intensity is almost 0). With more and more solvent added to the solution, PNIPAm started to become hydrophobic, agregating according to hydrophobic effect, results in the increase of scattering intensity. This indicates that more and more crumpled coil structures are formed (the wide standard deviation of sample's PDI indicates the unstable structure). When more solvent is added to the system, PDI of the samples go down, which means particles are uniform, compact PNIPAm globule structures are formed.

10. The sentence on lines 337-340 requires revision.

Answer: We thank the reviewer for the comments. We revised this part in the manuscript:

The larger size of PNIPAm particles of hundreds of nm in our experiment is another evidence. According to the above investigation we assume that the PNIPAm inserted into polymersomes membrane by slowly tangled but not compacted and appeared as crumpled coil (Figure 6b) in the membrane.

Reviewer #3 (Remarks to the Author):

This interesting contribution by Wilson and coworkers shows the shape transformation of polymer vesicles by introducing hydrophobic macromolecular guests to the block copolymer (BCP) bilayer. They used the LCST behaviour of PNIPAM in the presence of organic solvents in water, which lowers the LCST temperature of the thermosensitive polymer proportional to the volume fraction of organic solvents in water. The authors had their motivation to mimic the behavior of biomembranes in the presence of membrane-bound proteins, which are responsible for the change of the curvature of lipid bilayers. The introduction of hydrophobic nano-sized objects such as nanoparticles to the block copolymer bilayer has been utilized to implement functional elements to polymer vesicles. However, it is scarce to use this idea for the intentional shape transformation of polymer vesicles. I am enthusiastic about this paper because the results reported herein could open a new way to manipulate the shape of polymer vesicles rationally by introducing membrane-protein mimics. If realized, this can

be significant for understanding how the polymer membrane behaves under perturbation arising from guests. The expansion of the potential of polymer vesicles for nanoscience and drug delivery would be an excellent prospect as well. However, I believe that the presented ideas of this paper should be carefully understood and applied to explain the data presented in this paper in terms of the fundamentals of the shape change of polymer membranes. Let me elaborate on my opinions.

Polymer vesicles consist of block copolymer bilayers, which are conceptually similar to lipid bilayers. BCP bilayers, however, are different from lipid bilayers in terms of physical robustness arising from the high molecular-weight of BCPs. This difference has to be considered when the analogy between BCP bilayers and lipid bilayers as the translational motion of a BCP within the bilayer can be extremely limited. Earlier studies by Eisenberg and others showed a very low critical micelle concentration of block copolymer micelles and limited (virtually non-existent) molecular exchange between polymer micelles. I will summarize how polymer vesicle changes their shape in response to external stimuli with these in mind. With a given internal volume and the surface area of the BCP bilayer, polymer vesicles adopt a spherical morphology in solution. Polymer vesicles change their shape when (1) the volume of the internal compartment changes; (2) the surface area of the BCP bilayer changes.

As I pointed out, BCP bilayers lack fluidity because of the limited translational motion of BCPs in the bilayer. Some hydrophobic polymers such as polyisoprene and polyethylene, having low glass transition temperatures provide fluidity to the BCP bilayers if the molecular weight of these polymers is limited (Bates and coworkers). BCP bilayers built with the high glass transition temperature hydrophobic polymers such as polystyrene can be fluidized by plasticizing polystyrene with organic solvents. The shape transformation of polymer vesicles with plasticized membranes can change their shape from a sphere to a discoid and stomatocyte upon the decrease in the volume of the internal compartment caused by the osmotic ejection of organic solvents. Wilson and coworkers are the experts at the forefront in this field.

Answer: We thank the reviewer for the comments. We would like to share some of our future expectations from this work in several aspects. For example, it is believed that the shape of nanocarriers can influence the cellular uptake, however, researches about this topic has been restricted by the shapes that we can fabricate no matter from liposomes or polymersomes. With the method we have here, more non-axisymmetric nanocarriers can be fabricated, which will definitely help us to deeper the understanding of this relationship. In terms of nanomachines, our group is trying to develop nanomotors with controlled motion, these non-axisymmetric nanocarriers will be good models for it. Moreover, the Brownian motion of spherical particles has been observed and described by Stokes-Einstein equation, however the Brownian motion and motion of non-spherical and non-axisymmetric particles has not been studied and defined. We hope our new findings can also help to gain the knowledge of this field. With the idea from this paper, the hybrid systems, like protein-polymer systems, can also benefit from it. For example, modifying proteins with polymers contains amphipathic helices property might help proteins to anchor on nanoparticles. In terms of in vivo protein- cell interaction, the modification of amphipathic helices polymers on proteins might increase the efficacy.

In this paper, organic solvents (THF/dioxane) were introduced to the aqueous dispersion of spherical polymer vesicles in the presence of solubilized PNIPAM. The solvent molecules plasticize polystyrene membrane of polymer vesicles and destabilize PNIPAM to be hydrophobic; (3) infiltrate to the membrane to increase the internal compartment volume. Figure 1 shows that the initial result of

the shape transformation is discoid, resulting from the decrease of the volume decrease of the inner compartment. These EM images raise one question: is the presented process resulting from the increase or decrease of the internal volume?-----decrease of the inner volume?

Answer: We thank the reviewer for the question. In this method, the addition of organic solvent has already decreased the volume of the inner compartment. The presented process is based on this flat disk. There might be a slightly decrease in the volume, but this should not be the reason that causes the shape changing, which is proved by the control experiment from Figure S3. Extra experiment with PAA and PVP has also been done to further prove the specificity of PNIPAm.

Figure S3. Temperature control in the insertion of PNIPAm. PEG 2k with no LCST transition was used for comparison. The transition of PNIPAm happened when increases the temperature from 0 °C to 21 °C near LCST of PNIPAm. Sample with the addition of 100 μ g PEG2k at 0 °C with 23.07% organic solvent (a), and increase the temperature to 21 °C (b). With the addition of 100 μ g PNIPAm10k at 0 °C with 23.07% organic solvent (c), and increase the temperature to 21 °C (d).

The structures look flat in some of the EM images presented in this paper. The destabilized PNIPAm is collapsed to hydrophobic nanoparticles (presumably single-chain particles), which entered into the hydrophobic polystyrene part of the membrane. The authors claimed that this inclusion of nanoparticles perturbed the curvature of the membrane and led to the formation of the buds (tentacles) around the perimeter of the discoid. Considering the nature of BCP bilayers that involve the interdigitation of polymer chains (as described in Fig 1), the hydrophobic polymer chains most likely would form a highly entangled domain. When the hydrophobic nanoparticles of PNIPAm single chains entered into the BCP bilayer, this event might increase the volume fraction of the hydrophobic domain compared to the PEG domain. The increase of the hydrophobic domain's volume also might change the morphology of polymer vesicles. The morphological change would be lower curvature

structures like flat lamella. My next question is: are the structures after PNIPAM introduction vesicles or flat structures?

Answer: We thank the reviewer for the comments. The structures after PNIPAM introduction remained flat, however still hold a small internal volume, therefore not a flat lamella (confirmed by cryo-SEM images in Figure 2), as the introducing of PNIPAm was not enough to cause the further shape changing from a disk to stomatocyte. Firstly, the polymer length is much shorter than PEG-PS; secondly, the PNIPAm is not totally inserted into the entangled PS but partially embedded in between PEG, which can be proved by NMR detection. As shown in Figure S8, when polymersomes with PNIPAm inserted were dispersed in D₂O, the PNIPAm can be detected by ¹H NMR, as well as PEG but not PS. This suggest that PNIPAm is not entangled in the PS segments and maintains its fluidity.

It is claimed that incorporated PNIPAM chains were localized at the end-cap of the protrusion (tentacle). The authors showed that the small particle-like electron-rich dots were mostly found from the ends of the tentacles. However, I am not convinced that Cu²⁺ binds with the amides of PNIPAM. Even in the case of Cu-binding to collapsed PNIPAM chains, I am not sure that this technique can provide enough contrast in a single molecule level, which allows the trace of the location of PNIPAM chains in the membrane. Can the authors have a more effective method to prove this?

Answer: We thank the reviewer for the comments. Cu²⁺ is known to coordinate with 4 or 6 of amides, a single NIPAM monomer, or a short PNIPAM chain, or any other O-containing group can provide only a few sites for coordination of Cu²⁺, possess relatively low coordination numbers, which makes such complexes unstable and can be hardly visualized¹⁶. Here, multiple coordination together allows the visualization of PNIPAm (presented as black dots), since PNIPAm10k has in principle 10k amides. Although Energy-dispersive X-ray spectroscopy (EDS) could detect the Nitrogen from PNIPAM, detecting elements with low atomic numbers is unreliable. Here, we used another method to visualize PNIPAm based on biotin-avidin interaction. Avidin is a protein derived from both avians and amphibians that shows considerable affinity for biotin, this avidin-biotin complex is the strongest known non-covalent interaction ($K_d = 10^{-15}M$) between a protein and ligand. First, biotin was linked on PNIPAm, then the PNIPAm-biotin was used to insert into polymersome membrane. After washing for several times, these polymersomes were mixed with Streptavidin modified Gold nanoparticles. With the biotin-avidin interaction, Gold nanoparticles should be seen on the membrane where PNIPAm were inserted. As showing below, after the free gold nanoparticles were removed by blotting method, samples were checked under TEM. We can see the results are similar to what we observed with the Cu²⁺ staining method. However, as the biotin on PNIPAm might be covered by PEG shell and PNIPAm, this method is not as sensitive as expected. But we think the following results could be used to support our hypothesis.

The segregation of embedded PNIPAM chains within the polymersome membrane is highly unlikely when one considers the viscosity of the solvent-swollen polystyrene domain. This claim should be carefully made based on experimental facts, not solely based on TEM images.

Answer: We thank the reviewer for the comments. In terms of the fluidity, polymersomes are less fluid than both liposomes and cellular membranes². However, that does not mean the segregation can't happen among polymersome membrane, several researches have showed the possibility of polymer segregation and used the segregation to achieve their goals. So-Jung Park and co-workers¹⁷ have reported DNA-induced polymer segregation and DNA island formation in binary block copolymer assemblies, in which the DNA is linked to the PEO shell of the particles. In another research, Jan van Hest and co-workers¹⁰ also proved the lateral segregation after assembly, which result in a spontaneous domain formation. Furthermore, the addition of plasticizer can change the rigidity of polymer membrane tremendously, which also helps to increase the fluidity of the membrane^{13, 14}. However, in this work we mentioned the segregation of PNIPAm, which is a post-inserted polymer, that only interact with the membrane. This means that PNIPAm is not totally tangled in the membrane, and the segregation is possible. Further discussion about the redistribution of PNIPAm is added in the manuscript:

To investigate the details of curvature formation, samples were taken at different time points after the addition of plasticizing solvent. As shown in Figure 5, the tentacles observed previously did not form immediately. Interestingly, triangular oblate shapes appeared with 10 μg PNIPAm presented and polygonal oblate shapes appeared with 100 μg PNIPAm presented at 23.08% VFC (Figure 5b,f), meaning that the insertion of PNIPAm is random, with no selectivity. However, after 10 minutes arrangement, these shapes transited to one tentacle polymersomes and polymersomes with 3-4 tentacles (Figure 5c-d, g-h) respectively. This rearrangement of PNIPAm has been followed with the copper ion negative staining method. As shown in Figure S13, PNIPAm was first everywhere on the membrane at 0 min, and then moved to the protrusions of the membrane at 20 min.

Previous studies have shown that the local composition change of phospholipids bilayer can be induced by the change of external environment, such as temperature, pH and osmolality based on the high membrane fluidity^{3, 4}, which results in phase separation in liposome membrane⁵⁻⁷. In contrast,

polymersome membrane has low fluidity because of the higher molecular weight of block copolymers and chain-chain entanglements during the assembly⁸. Here, the amount of plasticizer added is also not enough for the reassembly of PEG-PS, due to the strong chain-chain entanglements⁹. Instead, the inserted PNIPAm is more fluid than PEG-PS. Firstly, the polymer length is much shorter than PEG-PS; secondly, the PNIPAm is not totally inserted into the entangled PS but partially embedded in between PEG, which is demonstrated by its detection via solution NMR. As shown in Figure S8, when polymersomes with PNIPAm inserted were dispersed in D₂O, the PNIPAm could be detected by ¹H NMR, as well as PEG but not PS. This suggests that PNIPAm is not entangled in the PS segments, therefore it maintains its fluidity. We assume the tentacle formation is mainly caused by the segregation of PNIPAm, resulting in its mobility along the membrane. The segregation of mixed polymer nanoscale vesicles has been considered to be difficult to achieve^{10, 11}, however, as a post-inserted polymer, PNIPAm has more freedom than PEG-PS, which allows it to segregate within the membrane. On the other hand, a lateral phase separation of the polymer is expected when there are mixed polymers, due to its low entropy of mixing. Similar segregation behaviour was also observed in liposomes by M. Kihner and co-workers⁷. Moreover, during the segregation, polymersomes' membrane has become thinner at the normal place but thicker at the protrusions, which kept the structure stable since the thick-walled compartments exhibit smaller spontaneous curvatures than the thin-walled compartment.

Figure S8 ¹H NMR spectra (400 MHz, 298 K) in D₂O of polymersomes, with free PNIPAm added (top), free PNIPAm (middle) and polymersomes with inserted PNIPAm (bottom). A broad signal from the PEG is observed from 4.0 to 3.0 ppm, together with a sharp peak at 3.63, representing PEG at and sticking out of the surface of the polymersomes respectively. When PNIPAm is added in polymersomes without solvent addition, it cannot enter the membrane and the signal of free PNIPAm can be observed (top). When PNIPAm is added to polymersomes followed with solvent addition, it can enter the membrane, and presented as solid state, since broad signals can be observed at 1.2, 0.8

and 0.0 ppm. The change of microenvironment might cause the peak shift. Meanwhile, the signal of free PNIPAm is not observed since its washed over (bottom). The broadness of the peaks indicate insertion in the membrane.

Figure S13. Staining of PNIPAm distribution in polymersome membrane. TEM image of polymersome with the addition of 100 μg PNIPAm at 23.08% VFC, and membrane dyed with copper sulfate (CuSO_4) at different time point (a) 0 min and (b) 20 min. Scale bar 200 nm. PNIPAm was first everywhere on the membrane, and then moved to the protrusions on the membrane.

When additional organic solvents were introduced to the polymersomes with embedded PNIPAM, the polymersomes underwent further shape changes from tubular structures to oblates and stomatocytes. This transition indicates that the internal volume of the polymersome must be reduced with the given surface area. Dose more organic solvents induce osmotic swelling of the inner compartment by introducing more solvent molecules? Could the shape change happen upon quenching the self-assembly? The extraction of embedded PNIPAM with additional organic solvents also has to be proved by experiments.

Answer: We thank the reviewer for the comments. During the experiment, the volume decrease is mostly controlled by the amount of organic solvent. The increase of membrane permeability allows inner volume to have a further decrease to form stomatocytes as shown in Figure S 14 even without the addition of PNIPAM. The quenching procedure can not cause the shape change, we also had concern about this. After checking the sample by cyro-TEM without quenching, we are now confident that quenching does not induce such shape changing. To further prove the departure of PNIPAM, we have used both FRET pair (cy3-cy5) and cy3-quencher pair to follow the procedure. The explanation is added in the main text:

To examine the dissociation of PNIPAM we used fluorescence resonance energy transfer (FRET) technique. PS linked with Cyanine3 (cy3) was embedded in polymersomes during the assembly, while PNIPAM linked with Cyanine5 (cy5) was used to interact with the self-assembled polymersomes. The interaction between PNIPAM and PS segments can be demonstrated by following the fluorescence intensity change from cy5 while exciting cy3. As shown in Figure S17, Cy5 fluorescence emission spectra (640-700 nm) was obtained by excitation of Cy3 at 512 nm. After addition of 150 μl organic

solvent to the system, the PNIPAm-cy5 inserted into polymersome membrane. When cy3 (embedded in the polymersome membrane during self-assembly) was excited at 512nm, the emission of cy5 was also excited by the emission from cy3 (blue line). After addition of 300 μ l organic solvent to the system, the PNIPAm-cy5 was slowly dissociated from polymersome membrane. When cy3 was excited at 512nm, the emission of cy5 was slightly excited by the emission from cy3, but to a much lesser extent than in the case when 150 μ l organic solvent was added (green line). Furthermore, a cy3 fluorescence quencher-Black Hole Quencher 2 (BHQ2) linked to PNIPAm was also applied to follow the dissociation. Similar results were observed as shown in Figure S18, the fluorescence from cy3 was quenched when PNIPAm-BHQ2 was inserted into the membrane at 150 μ l organic solvent addition. However, after 300 μ l organic solvent was added to the system, cy3 fluorescence increased back to normal (no quencher), suggesting the dissociation of PNIPAm at higher VFC.

Figure S17. Donor (polymersomes-Cy3) to acceptor (PNIPAm-Cy5) fluorescence energy transfer (FRET) demonstrating during the PNIPAm-membrane interaction. Cy5 fluorescence emission spectra (640-700 nm) obtained by excitation of Cy3 at 512 nm. After addition of 150 μ l organic solvent to the system, the PNIPAm-cy5 inserted into polymersome membrane. When cy3 (embedded in the polymersome membrane during self-assembly) was excited at 512nm, the emission of cy5 was also excited by the emission from cy3 (blue line). After addition of 300 μ l organic solvent to the system, the PNIPAm-cy5 was slowly dissociated from polymersome membrane. When cy3 was excited at 512nm, the emission of cy5 was slightly excited by the emission from cy3, but much lesser than 150 μ l organic solvent addition (green line).

Figure S18, PNIPAm dissociation determined followed by cy3-BHQ2 using fluorescence spectroscopy. Cy3 fluorescence emission spectra obtained by excitation of Cy3 at 512 nm. After addition of 150 μ l organic solvent to the system, the PNIPAm-BHQ2 inserted into polymersome membrane, the excitation of cy3 was quenched by BHQ2. After addition of 300 μ l organic solvent to the system, the PNIPAm-BHQ2 was slowly dissociated from polymersome membrane the excitation of cy3 regained.

Additional comments are

The Δa values shown on page 14-15 seem unsubstantiated. Please provide how these numbers can be used.

Answer: We thank the reviewer for the comments. The calculation of surface area and vesicle volume for polymersomes has been discussed in our previous work¹², which is based on cylindrically symmetric polymersomes. In the current case, the protrusions formed from the membranes have made the calculations more difficult as most of the shapes we created are non-axisymmetric shapes. If we need to follow the experimental results, we have to actually cut the structure into pieces to know the exact parameters to calculate the volume, which is impractical as the structure is so small. At this moment, we are trying to find another calculation method, and hope to find a way to simulate the shape changes in this experiment. On the other hand, based on the simulation we have done before¹², we knew that the parameters in polymersomes' simulation is similar to liposomes in the same structures, which is why in this manuscript we compared the results we got with the liposome phase diagram. To make it easy to understand, we cited the $\Delta(a)$ from liposomes in the explanation. To make it clear, we added the reference in the manuscript.

Why was only 10k PNIPAM used for the experiments? How about PNIPAM with other molecular weight?

Answer: We thank the reviewer for the comments. We have also tried PNIPAM with other molecular weight, for example 2k, and 7k. For the shorter length PNIPAm 2k, we hardly observe the formation

of protrusions, for the 7k, we need a higher polymer concentration to form the same structure. It might be because shorter chains can hardly stay in the membrane as they are easier to diffuse.

Line 241, change Figure 10-11 to Figure S10-11.

Answer: We thank reviewer for the kind observation. We have corrected this in the main text.

In conclusion, we believe that the revised version has addressed the points raised by editor and the reviewers. We appreciate the help in improving the manuscript.

Reference

1. Ross, M. H.; Pawlina, W., *Histology*. Lippincott Williams & Wilkins: 2006.
2. Lee, J. C.-M.; Santore, M.; Bates, F. S.; Discher, D. E. From Membranes to Melts, Rouse to Reptation: Diffusion in Polymersome Versus Lipid Bilayers. *Macromolecules* **2002**, *35*, 323-326.
3. Farge, E.; Devaux, P. F. Shape Changes of Giant Liposomes Induced by an Asymmetric Transmembrane Distribution of Phospholipids. *Biophys. J* **1992**, *61*, 347-357.
4. Sackmann, E.; Duwe, H.-P.; Engelhardt, H. Membrane Bending Elasticity and Its Role for Shape Fluctuations and Shape Transformations of Cells and Vesicles. *Faraday Discuss. Chem. Soc.* **1986**, *81*, 281-290.
5. Yanagisawa, M.; Imai, M.; Taniguchi, T. Shape Deformation of Ternary Vesicles Coupled with Phase Separation. *Phys. Rev. Lett.* **2008**, *100*, 148102.
6. Baumgart, T.; Hess, S. T.; Webb, W. W. Imaging Coexisting Fluid Domains in Biomembrane Models Coupling Curvature and Line Tension. *Nature* **2003**, *425*, 821-824.
7. Simon, J.; Kühner, M.; Ringsdorf, H.; Sackmann, E. Polymer-Induced Shape Changes and Capping in Giant Liposomes. *Chem. Phys. Lipids* **1995**, *76*, 241-258.
8. Battaglia, G.; Ryan, A. J. Bilayers and Interdigitation in Block Copolymer Vesicles. *J. Am. Chem. Soc.* **2005**, *127*, 8757-8764.
9. Meeuwissen, S. A.; Kim, K. T.; Chen, Y.; Pochan, D. J.; van Hest, J. C. M. Controlled Shape Transformation of Polymersome Stomatocytes. *Angew. Chem. Int. Ed.* **2011**, *50*, 7070-7073.
10. Meeuwissen, S. A.; Bruekers, S. M.; Chen, Y.; Pochan, D. J.; van Hest, J. C. Spontaneous Shape Changes in Polymersomes Via Polymer/Polymer Segregation. *Polymer Chemistry* **2014**, *5*, 489-501.
11. Zhuang, Y.; Lin, J.; Wang, L.; Zhang, L. Self-Assembly Behavior of Ab/Ac Diblock Copolymer Mixtures in Dilute Solution. *J. Phys. Chem. B* **2009**, *113*, 1906-1913.
12. Rikken, R. S. M.; Engelkamp, H.; Nolte, R. J. M.; Maan, J. C.; van Hest, J. C. M.; Wilson, D. A.; Christianen, P. C. M. Shaping Polymersomes into Predictable Morphologies Via out-of-Equilibrium Self-Assembly. *Nature Communications* **2016**, *7*, 12606.
13. Wang, E.; Zhu, J.; Zhao, D.; Xie, S.; Bates, F. S.; Lodge, T. P. Effect of Solvent Selectivity on Chain Exchange Kinetics in Block Copolymer Micelles. *Macromolecules* **2020**, *53*, 417-426.
14. Discher, D. E.; Eisenberg, A. Polymer Vesicles. *Science* **2002**, *297*, 967.
15. Zihlerl, P.; Svetina, S. Nonaxisymmetric Phospholipid Vesicles: Rackets, Boomerangs, and Starfish. *EPL (Europhysics Letters)* **2005**, *70*, 690.
16. Sinha-Ray, S.; Zhang, Y.; Placke, D.; Megaridis, C. M.; Yarin, A. L. Resins with "Nano-Raisins". *Langmuir* **2010**, *26*, 10243-10249.
17. Luo, Q.; Shi, Z.; Zhang, Y.; Chen, X.-J.; Han, S.-Y.; Baumgart, T.; Chenoweth, D. M.; Park, S.-J. DNA Island Formation on Binary Block Copolymer Vesicles. *Journal of the American Chemical Society* **2016**, *138*, 10157-10162.

REVIEWER COMMENTS

Reviewer #2 (Remarks to the Author):

I thank the authors for their clarifications.

Based on them, I now understand that the deformation of the polymersomes is driven by the lateral phase segregation of the inserted PNIPAm molecules. The segregation is facilitated by the ability of the PNIPAm to diffuse in the polymer bilayer even though the PEG-PS molecules, forming the matrix of the bilayer, are strongly entangled and immobile. The segregation enhances locally the area difference, which is relieved by outward membrane deformation.

In the newly added paragraph in the main text (lines 272-296) I believe there are few statements that require revision or clarification. Once completed, the manuscript will be suitable for publication in Nature Communications.

1. "The inserted PNIPAm is more fluid than PEG-PS" is not a correct statement. Fluidity is a property of matter, but not of single molecules. Single molecules can locally diffuse, even if the surrounding membrane appears immobile at longer length scales (e.g. <https://doi.org/10.1016/j.cell.2018.09.054>)
2. "We assume the tentacle formation is mainly caused by the segregation of PNIPAm, resulting in its mobility along the membrane" (285-287). Segregation does not result in PNIPAm mobility, it results or is facilitated by the mobility of the molecules.
3. "Moreover, during the segregation, polymersomes' membrane has become thinner at the normal place but thicker at the protrusions, which kept the structure stable since the thick-walled compartments exhibit smaller spontaneous curvatures than the thin-walled compartment"
I don't understand this conclusion. Smaller spontaneous curvature would mean that the preferred shape of the membrane is a flat sheet. The fact that PNIPAm is inserted only in the upper leaflet and that it is enriched in the protrusions suggests to me that the protrusions are stabilised by the increased (not decreased) spontaneous curvature, generated by the PNIPAm molecules.

Reviewer #3 (Remarks to the Author):

The manuscript is now improved with additional discussions and experiments added in the manuscript and supporting information. Although I believe that the experimental method to alter the shape of polymersomes to the desired one in a rational fashion is a very important addition to the existing toolbox of nanoscience by the self-assembly of block copolymers, I feel that the current manuscript still has some issues in terms of the solid experimental supports of the claims raised by the authors.

1. The driving force of the inclusion of PNIPAM into the membranes of PEG-PS polymersomes would

be the hydrophobicity of destabilized PNIPAM. The results in this paper suggest that most of PNIPAM are included in the membrane. If this was true, PNIPAM would be embedded in the hydrophobic PS domain of the membrane. Although it is virtually impossible to quantify the degree of the inclusion of PNIPAM in the hydrophobic domain of the membrane, I would assume that the degree of inclusion should be high. This embedded domain induces the budding of the membrane, which suggests the increase of the curvature. This is contrary to the existing idea that the higher curvature area would have higher interfacial energy, which requires additional stabilization. It is difficult to think that the destabilized PNIPAM could provide stabilization at the higher interfacial area considering the membrane is stabilized by PEG corona. I would assume that the curvature could be lower when the PNIPAM is embedded in the BCP membrane.

2. The shape change of polymersomes described in this paper is first caused by the decrease of the volume of the internal compartment by the addition of organic solvents. This caused the shape change from spherical vesicles to discoids due to the reduced volume. The addition of PNIPAM to the membrane only increases the surface area of the membrane, which develops the tentacles. This is caused by the increase of Δa . From the studies on liposomes (ref. 16), the tentacle formation requires a substantial increase of Δa (without significant change of the reduced volume). The results in this paper suggest that a significant amount of added PNIPAM should be embedded in the membrane. The authors claimed in this paper that PNIPAM is only partially embedded in the membrane. This is puzzling. How does this partial inclusion of PNIPAM increase the value of Δa ?

3. Care must be taken for the interpretation of NMR data. Why the chemical shifts of the peaks assigned to the protons of embedded PNIPAM is significantly downshifted? Why the intensities of the peaks of embedded PNIPAM changed significantly? If the suggested Δa value to induce the tentacle formation (especially with a large amount of added PNIPAM), could the author quantify the amount of PNIPAM embedded in the membrane by measuring the NMR of the dissolved polymersomes in organic solvent? If PNIPAM was only partially embedded in the membrane, why the peaks from PNIPAM were not pronounced in the spectrum? Also, the NMR analysis requires the complete absence of free PNIPAM in the dispersion solution. How did the author make sure about this issue?

4. The desorption of the embedded PNIPAM from the membrane was not fully supported by experimental results. When the fraction of the organic solvents increased to lower the LCST of PNIPAM, the increased fraction of organic solvents would induce the volume change of the internal compartment due to the osmotic pressure. How did the authors conclude that the shape change was only caused by the desorption of PNIPAM? If the lowered LCST could be regarded as the increased hydrophobicity of PNIPAM, why was PNIPAM excluded from the hydrophobic membrane?

5. reference 36 showed that the high contrast in TEM came from CuO crystals. Was this the case in the paper? Where was the coordination site? Was it the exposed section of PNIPAM or embedded PNIPAM? I am still not convinced that the CuSO₄ staining of PNIPAM is a conclusive method to prove the location of PNIPAM in the membrane.

Reviewer #2 (Remarks to the Author):

I thank the authors for their clarifications.

Based on them, I now understand that the deformation of the polymersomes is driven by the lateral phase segregation of the inserted PNIPAm molecules. The segregation is facilitated by the ability of the PNIPAm to diffuse in the polymer bilayer even though the PEG-PS molecules, forming the matrix of the bilayer, are strongly entangled and immobile. The segregation enhances locally the area difference, which is relieved by outward membrane deformation.

In the newly added paragraph in the main text (lines 272-296) I believe there are few statements that require revision or clarification. Once completed, the manuscript will be suitable for publication in Nature Communications.

1. "The inserted PNIPAm is more fluid than PEG-PS" is not a correct statement. Fluidity is a property of matter, but not of single molecules. Single molecules can locally diffuse, even if the surrounding membrane appears immobile at longer length scales (e.g. <https://doi.org/10.1016/j.cell.2018.09.054>)

Answer: We thank the reviewer for the observation. We have revised the sentence accordingly in the manuscript as shown below:

Here, the amount of plasticizer added is also not enough for the reassembly of PEG-*b*-PS, due to the strong chain-chain entanglements¹. Instead, the inserted PNIPAm as single polymer chains can locally diffuse along the membrane².

2. "We assume the tentacle formation is mainly caused by the segregation of PNIPAm, resulting in its mobility along the membrane" (285-287). Segregation does not result in PNIPAm mobility, it results or is facilitated by the mobility of the molecules.

Answer: We thank the reviewer for the kind observation. We revised the sentence in the manuscript as below:

We assume the tentacle formation is mainly caused by the segregation of PNIPAm facilitated by the mobility of the polymer chains along the membrane.

3. "Moreover, during the segregation, polymersomes' membrane has become thinner at the normal place but thicker at the protrusions, which kept the structure stable since the thick-walled compartments exhibit smaller spontaneous curvatures than the thin-walled compartment"

I don't understand this conclusion. Smaller spontaneous curvature would mean that the preferred shape of the membrane is a flat sheet. The fact that PNIPAm is inserted only in the upper leaflet and that it is enriched in the protrusions suggests to me that the protrusions are stabilised by the increased (not decreased) spontaneous curvature, generated by the PNIPAm molecules.

Answer: We thank the reviewer for the kind observation. Indeed, we have made a typing mistake here, we agree with the reviewer's opinion, it should be an increased spontaneous curvature. We revised the sentence in the manuscript as below:

Moreover, during the segregation, polymersomes' membrane has become thinner at the normal place but thicker at the protrusions, which kept the structure stable since the thick-

walled compartments exhibit larger spontaneous curvatures than the thin-walled compartment.

Reviewer #3 (Remarks to the Author):

The manuscript is now improved with additional discussions and experiments added in the manuscript and supporting information. Although I believe that the experimental method to alter the shape of polymersomes to the desired one in a rational fashion is a very important addition to the existing toolbox of nanoscience by the self-assembly of block copolymers, I feel that the current manuscript still has some issues in terms of the solid experimental supports of the claims raised by the authors.

1. The driving force of the inclusion of PNIPAM into the membranes of PEG-PS polymersomes would be the hydrophobicity of destabilized PNIPAM. The results in this paper suggest that **most of PNIPAM** are included in the membrane. If this was true, PNIPAM would be **embedded in the hydrophobic PS domain** of the membrane. Although it is virtually impossible to quantify the degree of the inclusion of PNIPAM in the hydrophobic domain of the membrane, I would assume that the degree of inclusion should be high. This embedded domain induces the **budding of the membrane, which suggests the increase of the curvature**. This is contrary to the existing idea that the higher curvature area would have higher interfacial energy, which requires additional stabilization. It is difficult to think that the destabilized PNIPAM could provide stabilization at the higher interfacial area considering the membrane is stabilized by PEG corona. I would assume that the curvature could be lower when the PNIPAM is embedded in the BCP membrane.

Answer: We thank the reviewer for the comments. We need to clarify that the amount of PNIPAm inserted into the membrane is much lower than the amount we added in the solution (see section 3 - NMR measurements). We suppose only the PNIPAm that has interaction with the polymersome membrane (before adding organic solvent) can be inserted into the membrane. Furthermore, PNIPAM is partially embedded in the hydrophobic PS domain of the membrane. This can be observed from the NMR results, as the signal of isopropyl from PNIPAM is much smaller, broader, therefore more shielded, than it was supposed to be in the liquid state spectrum, which suggested that this segment is closer to the solid part of the membrane. We were able to quantify the PNIPAm by NMR, in the new version the calculation of the amount of PNIPAm is added in the supporting information. In comparison to PS segments, the amount of PNIPAm is much lower (0.17mg in 10 mg of PEG-PS).

Curvature formation by hydrophobic insertion is often occurring in natural processes, the BAR proteins induced curvature has been widely explained and studied³. Membrane curvature is generated to minimize the elastic energy of the membrane, whether the curvature would be lower or higher depends on the type of insertion. Shallow insertion of a helical peptide can induce positive membrane curvature (protrusions), whereas deeper insertion may induce membrane curvature in the opposite direction (negative curvature – endocytic type structures)^{4, 5}.

For the assembly of block copolymers, the obtained morphology not only results from geometrical aspects but also from the minimization of the interfacial energy of the hydrophobic-hydrophilic interface⁶. However, during PNIPAM partial insertion, as the curvature is generated and protrusions start to form, minimization of the interfacial energy would facilitate further insertion and lateral segregation of the much more mobile PNIPAM chains.

2. The shape change of polymersomes described in this paper is first caused by the

decrease of the volume of the internal compartment by the addition of organic solvents. This caused the shape change from spherical vesicles to discoids due to the reduced volume. The addition of PNIPAM to the membrane only increases the surface area of the membrane, which develops the tentacles. This is caused by the increase of Δa . From the studies on liposomes (ref. 16), the tentacle formation requires a substantial increase of Δa (without significant change of the reduced volume). The results in this paper suggest that a significant amount of added PNIPAM should be embedded in the membrane. The authors claimed in this paper that PNIPAM is only partially embedded in the membrane. This is puzzling. How does this partial inclusion of PNIPAM increase the value of Δa ?

Answer: We thank the reviewer for the comments. The volume decrease of the internal compartment by the addition of organic solvent and the increase of surface area should happen at the same time, without PNIPAM, the only change will be the reduced volume to form disk structures. In reference 16, the osmotic pressure was also applied by adding 2 mM of sorbitol, which led to a decrease of vesicle volume. In literature, the surface area difference change is always coupled with the reduced volume⁷, resulting in membrane bending and shape changing. In this manuscript, the reduced volume was induced mainly by the solvent exchange between the inner and outside of the membrane. The PNIPAM does not contribute to the induced osmotic pressure and is ~ 0 as measured by osmometer. Instead, PNIPAM interacts with the membrane and is contributing to the positive curvature.

We consider the PNIPAM is partially embedded, due to the much longer chain (10k) comparing to PEG (2k). It's also because of the longer length, PNIPAM can be inserted deeper and stayed in the membrane but not being washed away after quenching. As long as the PNIPAM is partially inserted in the membrane, the Δa is increased. As shown in Figure S1, three surface areas are defined in bilayer particles, the actual surface area A is determined by the neutral plane of the bilayer. A_{out} means the outer surface area determined by the neutral planes of the outer monolayer, and A_{in} means the inner surface determined by the inner monolayer. Area difference of polymersome between two monolayers (ΔA) is calculated by A_{out} minus A_{in} . The reduced monolayer area difference (Δa) is calculated by dividing the area difference ΔA of a vesicle by that of a sphere with the same surface area A . As shown in Figure S1, any change in A_{out} can cause the increase of the reduced monolayer area difference (Δa). The partially embedded PNIPAM can only change A_{out} but not the A_{in} , since its not totally inserted in the membrane. Therefore, the reduced monolayer area difference (Δa) was increased, which resulted in curvature formation.

Figure S1. Area difference of polymersome between two monolayers (ΔA). Three surface areas are defined in bilayer particles, the actual surface area A is determined by the neutral plane of the bilayer (red line). A_{out} means the outer surface area determined by the neutral planes of the outer monolayer, and A_{in} means the inner surface determined by the inner monolayer (blue dashed lines). The reduced monolayer area difference (Δa) is calculated by dividing the area difference ΔA of a vesicle by that of a sphere with the same surface area A ^{8,9}.

3. Care must be taken for the interpretation of NMR data. Why the chemical shifts of the peaks assigned to the protons of embedded PNIPAM is **significantly downshifted**? Why the **intensities of the peaks** of embedded PNIPAM changed significantly? If the suggested Δa value to induce the tentacle formation (especially with a large amount of added PNIPAM), could the author **quantify the amount of PNIPAM embedded** in the membrane by measuring the NMR of the dissolved polymersomes in organic solvent? If PNIPAM was only partially embedded in the membrane, **why the peaks from PNIPAM were not pronounced** in the spectrum? Also, the NMR analysis requires the **complete absence of free PNIPAM** in the dispersion solution. How did the author make sure about this issue?

Answer: We thank the reviewer for the questions regarding the NMR data. We have revised the text to better explain the interpretation of the data. We assigned the downfield signals to the PNIPAM because these peaks arose only when PNIPAM was added, compared to our control study without PNIPAM present, we also dissolved the same sample in $CDCl_3$, and the PNIPAM peaks were also found there. Together with the negative staining experiment showing PNIPAM on the membrane, we concluded the signals are most likely from PNIPAM. The significant downshift of the signal coming from the PNIPAM can be explained due to the change in the surrounding environment of the PNIPAM, the increased hydrophobicity induced by the packing of PNIPAM and the PEG shielding. We have quantified the amount of PNIPAM that was embedded in the polymersomes and have added this amount to the supporting information. We have also clarified in the article why PNIPAM peaks were not very pronounced in the spectrum and how excess of free PNIPAM was removed. In short, the polymersomes were washed until no free PNIPAM was observed on NMR. The PNIPAM

embedded in the membrane is in the solid state and is therefore not visible on liquid NMR. However, some parts of the PNIPAm stick out of the membrane partially due to the sudden quenching and longer chain. These parts are slightly more soluble, and therefore visible on liquid NMR, though in low intensity and broader peak. By diffusion NMR, we confirmed these signals belonged to particles with an extremely slow diffusion (polymersomes are much larger structures than the free polymer chains), which means, the PNIPAm is embedded in the polymersomes membrane and not free in solution. Upon resolution we were able to calculate the amount of PNIPAm present in the membrane. To make it clear, we revised the manuscript accordingly as shown below:

The presence of PNIPAm is demonstrated both in polymersome solution (Figure S7-8) and after dissolving samples in CDCl_3 (Figure S9). After ensuring all free PNIPAm was washed out, we found the signals from the inserted PNIPAm are significantly downshifted, broadened and less intense compared to the free PNIPAm. The downward shift of the signals could be due to increased hydrophobicity induced by the packing of PNIPAm and the PEG shielding. The signals are broad and small because most inserted PNIPAm is in the solid phase, which cannot be seen on liquid NMR. Only small parts of the PNIPAm chain on the surface are slightly more dissolved by the solvent, making them liquid-like, resulting in small and broad peaks^{10, 11}. The amount of PNIPAm was also quantified, as 0.16 mg was found in the membrane in 10mg of PEG-*b*-PS. Since PNIPAm is inserted in the polymersome membrane and measured in solution (D_2O), broad peaks were observed from the backbone as it is in a solid phase and is covered by the PEG layer. More evidence is also provided by diffusion NMR. Free PNIPAm and polymersome with free PNIPAm added in the solution showed similar diffusion coefficient (FigureS10-11), however, when PNIPAm is inserted in the membrane, almost no decay of the peak was observed, similar as PEG peak from the polymersomes, as the nanoparticles are too big to diffuse within the detection limit of the probe (Figure S12).

Figure S9 ^1H NMR spectra (400 MHz, 298 K) of PNIPAm_{10k}. After freeze-drying the polymersomes with PNIPAm 10k inserted and re-dissolving in CDCl_3 (bottom), and free PNIPAm 10k in CDCl_3 (top) for comparison. PEG-*b*-PS is predominantly observed, as it made up the membrane of the polymersome. A small amount of PNIPAm can also be observed by the signals from the branches (c and d). The backbone (a and b) is not clearly visible, as it overlaps with the backbone of the PS. The PNIPAm (10k Da, $n=89$) peak at 4.0 ppm integrates to 3.18 compared to the PEG-*b*-PS, indicating around 3.6% PNIPAm

compared to the molar amount of PEG-*b*-PS (10 mg, 0.45 μmol) in the sample. The total amount of PNIPAm inserted is 0.0162 μmol , 0.16 mg.

4. The desorption of the embedded PNIPAM from the membrane was not fully supported by experimental results. When the fraction of the organic solvents increased to lower the LCST of PNIPAM, the increased fraction of organic solvents would induce the volume change of the internal compartment due to the osmotic pressure. How did the authors conclude that the shape change was only caused by the desorption of PNIPAM? If the lowered LCST could be regarded as the increased hydrophobicity of PNIPAM, why was PNIPAM excluded from the hydrophobic membrane?

Answer: We thank the reviewer for the comments. There might be a misunderstanding about the conclusion we got from the results. It is true that the increased fraction of organic solvents not only lower the LCST of PNIPAM, but also induce the volume change of the internal compartment due to the osmotic pressure, which will lead to the shape transformation from polymersomes to stomatocytes in the situation when there is no PNIPAm added as shown in Figure S14. The desorption of PNIPAm is not a reason but a result during the process. However, the inserted PNIPAm do hindered the shape transformation, as shown in Figure 6a, the shape transformation happened after the PNIPAM slowly dissociated from the membrane (tentacles disappear and disks are formed). We were trying to find out where was the PNIPAm during the process, do they go deeper in the membrane or dissociate from the membrane? According to the fluorescence experiments (Figure S16-17), we observed the embedded PNIPAm at 23% of organic solvent ratio, which is similar to the NMR results, however, when more and more solvent was added to the system, the embedded PNIPAm was not present in the membrane anymore, which indicates the dissociation of PNIPAm during the process.

To find out why the dissociation happened, we investigated PNIPAm's behaviour during the process alone. It turned out that the PNIPAm can form particles when organic solvent is added in the system, the more organic solvent, the more stable the PNIPAm particles are. As the PNIPAm particles formation and PNIPAm-membrane interaction happen simultaneously, we believe PNIPAm would prefer to form PNIPAm particles. On the other hand, when PNIPAm is getting more and more hydrophobic the interaction with PEG can become more difficult, which hinders the further insertion of PNIPAm, as PNIPAm is much longer than PEG, a fully insertion could take much more energy, the part that is not inserted might prefer to get together with the free ones, which might result in the occurrence of dissociation. This is also the reason that the insertion can only happen when the PNIPAm is only slightly hydrophobic. To make it clear and easy to read, we have revised this part in the manuscript, as shown below:

The curvature in natural membranes are reversible, meaning that proteins can be dissociated and reused in a new cycle of the vesicle formation⁴⁸. To investigate the reversibility of our model, more organic solvent was added to the system with 250 μg PNIPAm addition. 3-armed starfish-like shape was observed with Δa around 1.6 at 23.07% VFC, and rackets-like shape and cigar-like shape ($\Delta a \sim 1.4/1.5$)¹⁶ appeared at 25.39% VFC (Figure 6a1-a2), followed by the appearance of two tentacles polymersomes with 27.63% VFC (Figure 6a3) and one tentacle polymersomes ($\Delta a \sim 1.2$)¹⁶ with 31.79% of VFC (Figure 6a4). When the VFC percentage reached 33.79%, polymersomes have transformed to oblates ($\Delta a \sim 1.0$) (Figure 6a5), and to stomatocytes ($\Delta a \sim 1.0$) at 37.5% of solvent (Figure 6a6). With the increase of the VFC, the shape changing ends up with stomatocytes, which is similar to the shape transformation without the addition of PNIPAm (Figure S14), meaning that there is almost no membrane composition difference between the two monolayers. This might suggest that the PNIPAm is slowly dissociated from the membrane while the membrane is becoming more and more flexible, the reversibility of the curvature can be

achieved by increasing the membrane flexibility. Moreover, this PNIPAm dissociation was also detected in other shapes with different PNIPAm concentrations (Figure S15). All the assemblies with various shapes in the presence of different concentrations of PNIPAm altered to stomatocytes shape at 37.5% VFC. To examine the dissociation of PNIPAm we used fluorescence resonance energy transfer (FRET) technique. PS linked with Cyanine3 (cy3) was embedded in polymersomes during the assembly, while PNIPAm linked with Cyanine5 (cy5) was used to interact with the self-assembled polymersomes. The interaction between PNIPAm and PS segments can be demonstrated by following the fluorescence intensity change from cy5 while exciting cy3. As shown in Figure S16, Cy5 fluorescence emission spectra (640-700 nm) was obtained by excitation of Cy3 at 512 nm. After addition of 150 μ l organic solvent to the system, the PNIPAm-cy5 inserted into polymersome membrane. When cy3 (embedded in the polymersome membrane during self-assembly) was excited at 512nm, the emission of cy5 was also excited by the emission from cy3 (blue line). After addition of 300 μ l organic solvent to the system, the PNIPAm-cy5 was slowly dissociated from polymersome membrane. When cy3 was excited at 512nm, the emission of cy5 was slightly excited by the emission from cy3, but to a much lesser extent than in the case when 150 μ l organic solvent was added (green line). Furthermore, a cy3 fluorescence quencher-Black Hole Quencher 2 (BHQ2) linked to PNIPAm was also applied to follow the dissociation. Similar results were observed as shown in Figure S17, the fluorescence from cy3 was quenched when PNIPAm-BHQ2 was inserted into the membrane at 150 μ l organic solvent addition (red line). However, after 300 μ l organic solvent was added to the system, cy3 fluorescence restored to normal (blue line), suggesting the dissociation of PNIPAm at higher VFC. We reasoned that the PNIPAm itself might have changed with the increase of VFC. On the other hand, the more and more flexible membrane gives PNIPAm more freedom to diffuse out of the membrane, as it's not anchored on the polymers. To prove this hypothesis, the behaviour of PNIPAm under the same condition was detected using dynamic light scattering (DLS) technique.

5. reference 36 showed that the high contrast in TEM came from CuO crystals. Was this the case in the paper? Where was the coordination site? Was it the exposed section of PNIPAM or embedded PNIPAM? I am still not convinced that the CuSO₄ staining of PNIPAM is a conclusive method to prove the location of PNIPAM in the membrane.

Answer: We thank the reviewer for the comments. It is known that CuO belongs to the monoclinic crystal system, in which the copper atom is coordinated by 4 oxygen atoms in an approximately square planar configuration^{12, 13}. As proved by A. L. Yarin and coworkers¹³, the Cu²⁺ can coordinate with PNIPAm and form CuO crystals. In principle, we used the same method to visualise PNIPAm, so it should be the same mechanism. According to literature, there should be coordination between amide group and Cu²⁺, which is also the coordination site for the formation of this complex¹⁴, as protonation of a neutral occurs at the amide oxygen, most of the metal ions complexation were formed here. The staining of PNIPAm was conducted after removing all the free PNIPAm. As shown in the picture below, the black dots are mostly present near the membrane, this doesn't mean that the staining only happened to the exposed PNIPAM. Instead, as both PEG_{2k} and PNIPAm_{10k} can't be visualised by electron microscopy due to the low contrast, this verified that the PNIPAm are mostly interacting with PEG.

The CuSO₄ staining is based on the staining method that has been widely used together with electron microscopy to visualise the biological samples and also polymers¹⁵. The staining method should be reliable. On the other hand, the coordination between Cu²⁺ and amide is also well-known, and the CuO crystal is known as black solid as well. And using the staining method we have seen the difference between samples with different amount PNIPAm inserted, the more PNIPAm, the more and the darker black crystal dots were visualised. If this method is not reliable, we shouldn't be able to observe the results. In addition, the staining of the membrane is not the only method we have used to verify the

insertion of the PNIPAM. The additional fluorescence experiments explained previously have also demonstrated the presence of PNIPAM and its insertion in the membrane.

In conclusion, we believe that the revised version has addressed the points raised by the reviewers. We appreciate the help in improving the manuscript.

1. Meeuwissen, S. A.; Kim, K. T.; Chen, Y.; Pochan, D. J.; van Hest, J. C. M. Controlled Shape Transformation of Polymersome Stomatocytes. *Angew. Chem. Int. Ed.* **2011**, *50*, 7070-7073.
2. Shi, Z.; Graber, Z. T.; Baumgart, T.; Stone, H. A.; Cohen, A. E. Cell Membranes Resist Flow. *Cell* **2018**, *175*, 1769-1779.e1713.
3. Simunovic, M.; Voth, G. A.; Callan-Jones, A.; Bassereau, P. When Physics Takes Over: Bar Proteins and Membrane Curvature. *Trends in Cell Biology* **2015**, *25*, 780-792.
4. Schmidt, N. W.; Wong, G. C. L. Antimicrobial Peptides and Induced Membrane Curvature: Geometry, Coordination Chemistry, and Molecular Engineering. *Current Opinion in Solid State and Materials Science* **2013**, *17*, 151-163.
5. Campelo, F.; McMahon, H. T.; Kozlov, M. M. The Hydrophobic Insertion Mechanism of Membrane Curvature Generation by Proteins. *Biophysical Journal* **2008**, *95*, 2325-2339.
6. Le Meins, J. F.; Sandre, O.; Lecommandoux, S. Recent Trends in the Tuning of Polymersomes' Membrane Properties. *The European Physical Journal E* **2011**, *34*, 14.
7. Seifert, U. Configurations of Fluid Membranes and Vesicles. *Advances in Physics* **1997**, *46*, 13-137.
8. van Rhee, P. G.; Rikken, R. S. M.; Abdelmohsen, L. K. E. A.; Maan, J. C.; Nolte, R. J. M.; van Hest, J. C. M.; Christianen, P. C. M.; Wilson, D. A. Polymersome Magneto-Valves for Reversible Capture and Release of Nanoparticles. *Nat. Commun.* **2014**, *5*, 5010.
9. Zihlerl, P.; Svetina, S. Nonaxisymmetric Phospholipid Vesicles: Rackets, Boomerangs,

and Starfish. *EPL* **2005**, *70*, 690-696.

10. Pylypchuk, I. V.; Lindén, P. A.; Lindström, M. E.; Sevastyanova, O. New Insight into the Surface Structure of Lignin Nanoparticles Revealed by 1h Liquid-State Nmr Spectroscopy. *ACS Sustainable Chemistry & Engineering* **2020**, *8*, 13805-13812.
11. Jiang, F.; Dallas, J. L.; Ahn, B. K.; Hsieh, Y.-L. 1d and 2d Nmr of Nanocellulose in Aqueous Colloidal Suspensions. *Carbohydrate Polymers* **2014**, *110*, 360-366.
12. Forsyth, J. B.; Hull, S. The Effect of Hydrostatic Pressure on the Ambient Temperature Structure of CuO. *Journal of Physics: Condensed Matter* **1991**, *3*, 5257-5261.
13. Rulíšek, L. r.; Vondrášek, J. Coordination Geometries of Selected Transition Metal Ions (Co²⁺, Ni²⁺, Cu²⁺, Zn²⁺, Cd²⁺, and Hg²⁺) in Metalloproteins. *Journal of Inorganic Biochemistry* **1998**, *71*, 115-127.
14. Sigel, H.; Martin, R. B. Coordinating Properties of the Amide Bond. Stability and Structure of Metal Ion Complexes of Peptides and Related Ligands. *Chemical Reviews* **1982**, *82*, 385-426.
15. Trent, J. S.; Scheinbeim, J. I.; Couchman, P. R. Ruthenium Tetraoxide Staining of Polymers for Electron Microscopy. *Macromolecules* **1983**, *16*, 589-598.

REVIEWERS' COMMENTS

Reviewer #2 (Remarks to the Author):

My concerns have been addressed by the authors and I am happy for the manuscript to be published.

Reviewer #3 (Remarks to the Author):

The technical points are now cleared mostly. I believe that the revised manuscript is now improved enough to be published in Nature Communications. I anticipate that this report would be widely accepted by researchers in the field of self-assembly of block copolymers, which could result in the rational manipulation of the shape of soft nanostructures such as polymersomes.

REVIEWERS' COMMENTS

Reviewer #2 (Remarks to the Author):

My concerns have been addressed by the authors and I am happy for the manuscript to be published.

Reviewer #3 (Remarks to the Author):

The technical points are now cleared mostly. I believe that the revised manuscript is now improved enough to be published in Nature Communications. I anticipate that this report would be widely accepted by researchers in the field of self-assembly of block copolymers, which could result in the rational manipulation of the shape of soft nanostructures such as polymersomes.

We appreciate the help in improving the manuscript from the reviewers.